# Think-at-Hard:
# Selective Latent Iterations to Improve Reasoning Language Models

Tianyu Fu [* 1 2]  Yichen You [* 1]  Zekai Chen [1]  Guohao Dai [3 2]  Huazhong Yang [1]  Yu Wang [1]

## Abstract

Improving the reasoning abilities of Large Language Models (LLMs), especially under parameter constraints, is crucial for real-world applications. Looped transformers address this by performing multiple latent iterations to refine each token beyond a single forward pass. However, we identify a *latent overthinking* phenomenon: most token predictions are already correct after the first pass, but are sometimes revised into errors in later iterations. We ask whether *selectively skipping* latent iterations can *improve accuracy*, and reveal significant potential with an oracle iteration policy that boosts performance by up to 7.3%. Motivated by this, we propose Think-at-Hard (TaH), a looped transformer optimized for selective iteration. TaH employs a lightweight neural decider to trigger latent iteration, only at tokens likely to be incorrect after the standard forward pass. During latent iterations, depth-aware Low-Rank Adaptation (LoRA) modules shift the objective from general next-token prediction to focused hard-token refinement. A duo-causal attention mechanism extends attention from the token sequence dimension to an additional iteration depth dimension, enabling cross-iteration information flow with full sequential parallelism. Experiments on nine benchmarks show consistent gains across math, QA, and coding tasks. With identical parameter counts, TaH outperforms always-iterate baselines by 3.8-4.4% while skipping iterations on 93% of tokens, and exceeds single-iteration Qwen3 baselines by 3.0-3.8%. When allowing $< 3\%$ more parameters from LoRA and decider, the gains further increase to 5.3-6.2% and 6.1-6.8%, respectively. Our code is available here.

---
[*]Equal contribution   [1]Tsinghua University [2]Infinigence AI [3]Shanghai Jiao Tong University. Correspondence to: Yu Wang <yu-wang@tsinghua.edu.cn>.

*Proceedings of the 43rd International Conference on Machine Learning*, Seoul, South Korea. PMLR 306, 2026. Copyright 2026 by the author(s).

## 1. Introduction

Recent advances in Large Language Model (LLM) reasoning have enabled broad applications across diverse domains (Jaech et al., 2024; Guo et al., 2025; Yang et al., 2025). With hundreds of billions of parameters, LLMs can generate complex Chain-of-Thought (CoT) to solve challenging tasks. At the same time, smaller language models have also drawn increasing attention. With only a few billion parameters, they offer compelling alternatives: lower costs, faster inference, and suitability for edge computing (Abdin et al., 2024; Team et al., 2025; Wang et al., 2025a).

At this crossroads, enhancing reasoning capabilities under parameter constraints becomes a central challenge. A common approach is to distill smaller models to mimic LLM CoT trajectories. However, not all tokens are equally predictable: certain tokens encode critical logic or reasoning directions that are fundamentally harder to predict (Lin et al., 2024; Fu et al., 2025a; Wang et al., 2025b). With limited computation per output token, small models quickly hit a performance ceiling and mispredict some of these tokens. Once critical errors occur, the reasoning trajectory can irrecoverably diverge, yielding drastically different outcomes.

Prior work proposes looped transformers to address this parameter–performance paradox (Hutchins et al., 2022; Saunshi et al., 2025; Zeng et al., 2025; Zhu et al., 2025). Instead of verbalizing the next token immediately after one forward pass, these models typically feed the last-layer hidden states back into the LLM for additional passes, refining representations in the latent space. After certain iterations, the final hidden states pass through the language modeling head to generate the next token. By uniformly scaling up iterations per token, these models can correct initially wrong token predictions, potentially increasing performance without increasing parameter count.

However, we identify a *latent overthinking* problem in looped transformers, where excessive iterations revise correct answers into wrong ones. As shown in Figure 8, while the second iteration corrects 8.7% of predictions, it also flips 2.1% of correct ones into errors. This occurs because most tokens, such as coherence or suffix tokens, are already predicted correctly after the first pass; further iterations may

*Figure 1.* Latent iterations can fix wrong predictions, but can also *overthink* and flip correct ones. *Selective* iteration only when needed can improve reasoning with reduced computation.

instead introduce harmful changes. This mirrors overthinking in explicit CoT reasoning (Wu et al., 2025), where additional reasoning steps degrade rather than improve answers. This reveals a surprising opportunity:

*selectively skipping latent iterations on most tokens can further increase model performance.*

We validate this with an oracle policy that iterates only on initially mispredicted tokens, as shown in Table 1. Compared to always iterating, this selective oracle can achieve up to 32% higher accuracy with an optimized architecture.

Achieving selective latent iteration presents three main challenges. First, the model architecture should enable cross-depth attention, allowing each iteration to access full context. This is crucial because when early tokens skip deeper iterations, later tokens must still access their representations from shallower depths. Meanwhile, this cross-depth flow cannot compromise the sequence-level parallelism essential for efficient training and prefilling. Second, the model must adapt to distribution shifts across iterations, while maximizing parameter reuse. Third, training must remain stable despite tightly coupled dependencies: the iteration policy depends on prediction quality at each depth, while that quality depends on the depths to which previous tokens are assigned by the iteration policy.

To address these challenges, we propose TaH, a looped transformer optimized for selective latent iteration. As shown in Figure 2, TaH employs a neural decider to determine whether to iterate or verbalize each token. We design a duo-causal attention mechanism to enable cross-depth attention with full sequence parallelism. To specialize deeper iterations for current-token refinement and preserve strong first-pass predictions, we apply LoRA adapters solely at iterations $d > 1$. We train TaH stably by aligning both the LLM backbone and iteration decider with a static oracle iteration policy. We summarize our contributions as follows.

- **Selective Latent Iteration**. We identify the latent overthinking phenomenon and quantify its influence on token prediction accuracy and downstream tasks. This insight motivates new directions where latent iteration is applied selectively to a few tokens for both better reasoning quality and efficiency.

- **Specialized Model Architecture**. We develop a model architecture that natively supports selective iterations. The dedicated duo-causal attention mechanism, LoRA adapters, and iteration decider enable efficient cross-depth information flow, objective transitions, and dynamic depth selection.

- **Stable Training**. We introduce a stable training scheme that uses a static oracle policy to decouple model adaptation and policy learning. It overcomes the circular dependency between iteration decisions and prediction quality.

We fine-tune TaH from Qwen3-Base 0.6B and 1.7B on Open-R1, then test it across nine reasoning benchmarks spanning math, QA, and coding tasks. TaH achieves average accuracy gains of 3.0% and 3.8% over standard single-iteration baselines. Compared to looped transformer Ouro (Zhu et al., 2025) trained with same data, TaH achieves 3.8-4.4% gains while reducing latent iterations by 93%.

## 2. Related Work

Unlike standard LLMs that verbalize at every autoregressive step, latent thinking shifts part of generation away from explicit natural-language CoT in order to improve reasoning (Li et al., 2025).

**Signal-guided control**. These methods keep reasoning in token space but steer computation by inserting control tokens. They add filler tokens (e.g., dots) (Pfau et al., 2024) and learnable [PAUSE] tokens (Goyal et al., 2024; Kim et al., 2025) for extra compute during decoding. They are lightweight but constrained to discrete-token interventions with limited latent control.

**Latent optimization**. These methods perform autoregressive reasoning directly in internal representations, emitting little or no intermediate text. They distill CoT into continuous embeddings via progressive replacement (Hao et al., 2024; Cheng & Van Durme, 2024), hidden-state alignment (Su et al., 2025; Liu et al., 2024), or logit-weighted embeddings (Zhang et al., 2025b). While efficient, these methods sacrifice interpretability, with training-based ones requiring heavy mitigation from verbal LLMs.

**Looped transformers**. These methods interleave latent and verbal reasoning, adding latent iterations before each token verbalization. Previous work focuses on scaling up iteration depths, with the main architectural focus on next-iteration inputs: reusing hidden states directly (Saunshi et al., 2025; Geiping et al., 2025; Zhu et al., 2025) or using logit-weighted embeddings (Zeng et al., 2025). Looped transformers achieve deeper computation without parameter increases. However, uniform depth scaling burdens training and inference, and risks overthinking already-correct tokens.

**Positioning**. TaH belongs to the looped transformer family, but identifies *selective iteration* as a new design principle to improve performance. While concurrent works (Bae et al., 2025; Zhu et al., 2025) also enable dynamic recursion, they degrade performance at non-maximum depths and require full retraining. TaH instead leverages existing pre-trained models, adding depth-aware LoRA and duo-causal attention to improve reasoning with minimal fine-tuning overhead.

## 3. Preliminaries

**Autoregressive LLMs**. Modern LLMs generate text through an autoregressive next-token prediction process. It includes a *prefill* stage and a *decode* stage (Radford et al., 2018; 2019; Kwon et al., 2023). In the prefill stage, the model processes the entire input sequence in parallel; in the decode stage, it consumes one new token at a time along with cached history to predict the next token.

Formally, let $t_i$ denote the token at position $i$ and $x_i \in \mathbb{R}^h$ its embedding. Let $E \in \mathbb{R}^{v \times h}$ be the embedding matrix, so $x_i = E[t_i]$ when $t_i$ is treated as an index. Here, $v$ and $h$ are the vocabulary size and hidden dimension. The output projection matrix is $W_{\text{out}} \in \mathbb{R}^{h \times v}$ (equal to $E^\top$ if tied). Given the context $T_{\leq i} = [t_0, \dots, t_i]$ with embeddings $X_{\leq i} = [x_0, \dots, x_i]$, the model $\theta$ produces a *last-layer hidden state* $y_i$ for token $t_i$:

$$y_i = \mathcal{P}_\theta(x_i \mid X_{\leq i}) \in \mathbb{R}^h. \tag{1}$$

The next-token distribution $p_i$ and sample are:

$$p_i = \text{softmax}(W_{\text{out}}^\top y_i) \in \mathbb{R}^v, \qquad t_{i+1} = \mathcal{S}(p_i), \tag{2}$$

where $\mathcal{S}$ is a sampling rule such as nucleus sampling. Decoding repeats until an end-of-sequence token is generated.

**Causal attention**. Modern LLMs typically adopt a *causal* attention mechanism. As shown in Figure 2(a), each position attends only to itself and earlier positions, consistent with Equation 1. This design brings two key benefits: (1) it enables parallel training with next-token prediction and shifted logits, avoiding the need for token-by-token generation; and (2) it allows efficient inference by caching Key-/Value states of past tokens instead of recomputing them.

**Looped transformers**. Looped transformers introduce an inner loop that iterates in latent space before verbalizing each output token. Let $d \in \{1, 2, \dots\}$ denote the iteration depth (written as a superscript), and set $x_i^{(0)} = E[t_i]$. At each iteration, looped transformers update $y_i$ with causal attention on the hidden states of *the current iteration*:

$$y_i^{(d)} = \mathcal{P}_\theta(x_i^{(d)} \mid X_{\leq i}^{(d)}), \qquad X_{\leq i}^{(d)} = [x_0^{(d)}, \dots, x_i^{(d)}]. \tag{3}$$

An inner transition then produces the next-depth embedding. For example, Loop (Saunshi et al., 2025) simply sets

*Table 1.* Performance comparison across iteration strategies, using Ouro-1.7B model except for the last row. Always1 means no latent iteration. Oracle policy iterates on 12-19% of tokens. MMLU100 denotes the first 100 questions from MMLU-STEM.

| Iter. Policy | NTP | AMC23 | MMLU100 | HE++ |
|---|---|---|---|---|
| Always1 | 73.1 | 38.1 | 56.0 | 39.6 |
| Always2 | 79.7 | 40.6 | 60.0 | 40.9 |
| Orcl. | 81.8/ + 2.1 | 47.9/ + 7.3 | 62.0/ + 2.0 | 43.3/ + 2.4 |
| Orcl. w.TaH | 89.3/ + 9.6 | 68.8/ + 28.2 | 85.0/ + 25.0 | 72.9/ + 32.0 |

$x_i^{(d+1)} = y_i^{(d)}$, while Ponder (Zeng et al., 2025) uses logit-weighted embeddings:

$$x_i^{(d+1)} = \text{softmax}(W_{\text{out}}^\top y_i^{(d)}) \, E = p_i^{(d)} E. \tag{4}$$

In practice, it uses the top-100 logits instead of full logits for efficiency.

Verbalization occurs at a fixed *maximum depth* $d_{\max}$ shared by all tokens, where $y_i^{(d_{\max})}$ is transformed into the next token $t_{i+1}$, resembling Equation 2.

## 4. Selective Latent Iteration Oracles

**Setup**. We analyze the latent iteration behavior of the Ouro looped transformer (Zhu et al., 2025) and our proposed TaH model (architecture detailed in Section 5). All models are fine-tuned from Qwen3-1.7B-Base on a balanced 100K-sample subset of the Open-R1 dataset (Hugging Face, 2025), following the setups in Section 6.1.

**Oracle iteration policy**. To investigate the potential of selective iteration, we establish an oracle policy $\pi$. It triggers additional iterations only when the reference LLM $\theta$ mispredicts the target token at the first forward pass. In this section, we set $\theta$ to the model under evaluation, so $\pi$ acts as a greedy, locally optimal policy.

Formally, let $\hat{p}_i$ denote the reference model's first-pass next-token distribution at position $i$, and $t_{i+1}$ the ground-truth token. The oracle iteration depth $d_i^\pi$ is:

$$d_i^\pi = 1 + \mathcal{D}(\hat{p}_i^{(1)}, t_{i+1}), \tag{5}$$

where $\mathcal{D}$ is a binary discrepancy metric. We use top-1 mismatch here: $\mathcal{D} = \mathbf{1}[\hat{t}_{i+1} \neq t_{i+1}]$, with $\hat{t}_{i+1} = \arg\max_t \hat{p}_i^{(1)}$ ($\mathbf{1}[\cdot]$ denotes the indicator function). For simplicity, we assume $d_{\max} = 2$. The arbitrary-depth case is in Appendix A.1.3, and alternative discrepancy metrics are ablated in Table 4.

For brevity, we call a token *easy* if the reference model correctly predicts it at the first pass ($d_i^\pi = 1$), and *hard* otherwise. Prior work (Fu et al., 2025a) shows that *hard* tokens typically occupy only a small proportion (e.g., 7%) of all tokens.

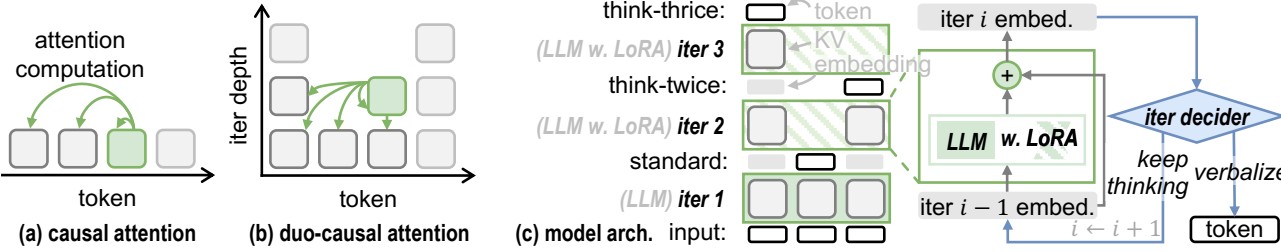

*Figure 2.* TaH Overview. (a) Regular causal attention: tokens attend only to previous positions. (b) Our duo-causal attention: tokens attend to both previous positions and shallower iteration depths, maintaining 2D causality. (c) Model architecture: TaH selectively iterates or verbalizes tokens. It uses LoRA at deeper iterations to shift from next-token prediction to hard-token refinement. A neural decider determines whether to continue iterating or output the token.

**Next-token prediction**. We evaluate next-token prediction (NTP) accuracy by comparing top-1 predictions against ground-truth tokens in the Open-R1 validation set. As shown in Table 1, always iterating twice improves NTP accuracy for Ouro by 6.6%, but at the cost of doubled depth. Surprisingly, the oracle policy, which skips iterations on 81-88% of tokens, further improves accuracy by 2.1%. This gain comes from avoiding *latent overthinking*: without selective iteration, the model can revise correct predictions to wrong ones at the second pass, as shown in Figure 8.

**Downstream tasks**. We next examine whether NTP gains translate to downstream task performance. Since ground-truth tokens are unavailable during generation, we use top-1 predictions from the stronger Qwen3-8B as proxy labels. Table 1 shows that the oracle policy improves downstream performance by 2.0-7.3%.

**TaH design objectives**. The oracle experiments reveal two key insights. First, models have significant untapped potential when iteration depth is selected correctly. Second, the oracle policy is effective enough to learn from, even though other globally optimal policies may exist. These findings motivate the design objectives of TaH: (1) its model architecture should natively support selective iteration depths; and (2) following the oracle policy, its training objective for deeper iterations is not to cover all tokens, but to selectively focus only on the few failing tokens. We later validate that TaH-1.7B can better utilize the oracle policy to achieve $> 25\%$ improvement, surpassing even Qwen3-4B. While the oracle requires ground-truth tokens unavailable at inference, approximating it via neural networks is a promising direction.

## 5. TaH Design

We expand the key motivations and designs of TaH in this section, including the duo-causal attention mechanism for cross-depth information flow (Section 5.1), the depth-adaptive model architecture with LoRA adapters (Section 5.2), and a two-stage training scheme (Section 5.3).

### 5.1. Duo-Causal Attention

**Motivation**. In looped transformers with fixed depth, standard causal attention on the current iteration's KV states incorporates all context (Equation 3). However, dynamic iteration depths create a challenge: tokens at deeper levels cannot access hidden states of previous tokens that verbalized at shallower depths. This poses a dilemma between requiring up-to-date context from all previous tokens and maintaining parallel training where depth-$d$ computations cannot depend on uncomputed deeper states ($d' > d$). Existing approaches compromise on one of these aspects. Some sacrifice parallelism by allowing attention to deeper iterations (Hao et al., 2024); others preserve parallelism by restricting attention to only the initial iteration's KVs (Bae et al., 2025). To resolve this dilemma, we introduce a simple yet effective mechanism to maximize cross-depth information flow while maintaining high parallelism.

**Duo-causal attention mechanism**. As shown in Figure 2(b), duo-causal attention extends *causality* to two dimensions, letting tokens attend across both previous positions and shallower iteration depths. Formally, we extend the accessible set from Equation 3 to

$$X_{\leq i}^{(\leq d)} = \{ x_j^{(k)} \mid j \leq i,\ k \leq d \}. \tag{6}$$

When all tokens iterate only once (as in standard transformers), this reduces to regular causal attention. The duo-causal design achieves both full parallel training and cross-depth information flow. At depth $d$, all tokens compute their depth-$d$ representations simultaneously using *only and all* information from depths 1 through $d$.

Duo-causal attention is fully compatible with attention kernels like FlashAttention (Dao et al., 2022; Dao, 2024; Shah et al., 2024) and sparse implementations (Fu et al., 2025b; Zhang et al., 2025a). As shown in Figure 3, we simply maintain separate KV caches per iteration depth and flatten the 2D (token, depth) grid into 1D by concatenating deeper KV caches after shallower ones. Positional encodings are applied based solely on the original token index, invariant to

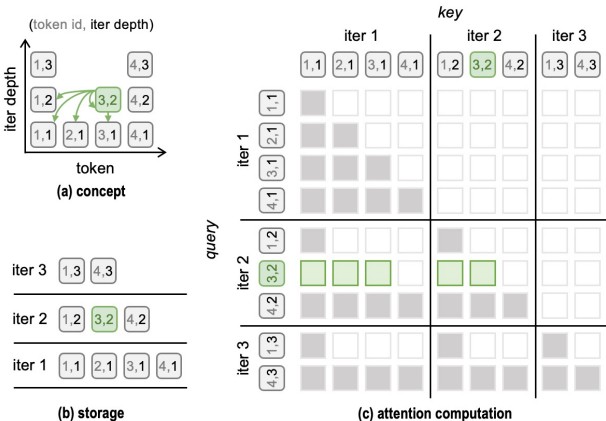

*Figure 3.* Duo-causal attention implementation. (a) Conceptual TaH example with dynamic iteration depths. Each cell denotes a token–depth pair (token id, iter depth). (b) Each iteration maintains its own KV cache. (c) KV caches from all iterations are concatenated into a 1D sequence and processed with standard attention under a duo-causal mask. The duo-causal mask is conceptually partitioned into blocks by iteration depth. The diagonal blocks use a standard causal mask, while off-diagonal blocks use reduced causal masks that enforce the duo-causal rules.

iteration depth. The duo-causal constraint is then enforced via a modified additive attention mask, requiring no custom CUDA kernels. More details on duo-causal attention implementation are discussed in Appendix A.4.1.

## 5.2. Depth-adaptive Model Architecture

**Motivation**. Previous looped transformers typically use identical weights across all iterations. However, we find that over 73% of next-tokens are correctly predicted at the first iteration (Table 1). This suggests deeper iterations serve a different objective: they refine the first iteration's prediction rather than predicting further ahead to the next-next token. This mirrors deep LLMs, where shallow layers predict next tokens for deeper layers to refine (Belrose et al., 2023; Schuster et al., 2022; Bae et al., 2023). While deep LLMs naturally handle this shift through distinct parameters per depth, looped transformers must accommodate both objectives with shared weights, potentially limiting performance. Moreover, fixed iteration depths can cause *latent overthinking*, motivating our dynamic approach.

**Backbone model**. To address the objective shift, we apply a LoRA adapter (Hu et al., 2022) to the shared LLM backbone only for iterations $d > 1$. As shown in Figure 2(c), this allows the base LLM to focus on latent embeddings, while the adapter handles the objective shift. It preserves strong next-token prediction at $d = 1$, alleviating interference from deeper iterations. We also add residual connections across iterations to simplify the refinement and improve gradient

flow. Formally, at depth $d$, we compute

$$y_i^{(d)} = \mathcal{P}_{\theta_d}\left(x_i^{(d)} \mid X_{\leq i}^{(\leq d)}\right), \quad (7)$$

with depth-specific parameters

$$\theta_d = \theta \text{ for } d = 1, \qquad \theta_d = \theta + \Delta \text{ for } d > 1,$$

where $\theta$ and $\Delta$ denote the LLM and LoRA weights, respectively. The next-iteration inputs use logit-weighted embeddings (Equation 4); verbalization follows standard sampling (Equation 2). Each $y_i^{(d)}$ either continues iterating or verbalizes according to the decider $\mathcal{I}_\phi$.

**Iteration decider**. We use a lightweight MLP as the iteration decider $\mathcal{I}_\phi$ to determine whether each token should continue iterating or verbalize. After each iteration, it concatenates the backbone's shallow, middle, and final hidden states to predict a continuation probability:

$$\hat{c}_i^{(d)} = \mathcal{I}_\phi(h_i^{(d)}) \in [0, 1].$$

During inference, token $i$ verbalizes when $\hat{c}_i^{(d)}$ falls below threshold $c_{\text{threshold}}$ or reaches maximum depth $d_{\max}$.

## 5.3. Training Scheme

We adopt a two-stage training scheme. We first fine-tune the backbone model to support selective latent iteration, then train the iteration decider. Both stages are aligned to the same oracle iteration policy.

**Motivation**. As shown in Figure 2(c), the backbone LLM and the iteration decider are tightly coupled, making joint training unstable. Specifically, the backbone's prediction quality across iterations determines the optimal depth, while the decider controls the backbone's KV cache and output depth. To stabilize training, we train the two components sequentially under a fixed oracle policy $\pi$ (validated in Section 4).

**Oracle Iteration Policy $\pi$.** Our goal is to iterate only on *hard* tokens that a standard supervised fine-tuned (SFT) model would mispredict on the first pass. Thus, we define $\pi$ with the SFT model as the reference: we trigger an additional iteration when its top-1 next-token prediction differs from the ground-truth token. In principle, one could instead define $\pi$ using the current looped model itself (i.e., an on-policy oracle), but we find this to be empirically unstable (Section 6.3).

**Stage 1: Backbone supervision under $\pi$.** We optimize the backbone LLM ($\theta$ and LoRA adapter $\Delta$) with $\pi$-guided iteration execution. The loss is standard next-token prediction at the oracle-determined depth:

$$\mathcal{L}_{\text{SFT}}(\theta, \Delta) = \sum_i -\log p_i^{(d_i^\pi)}(t_{i+1}),$$

where $p_i^{(d_i^\pi)}$ is the next-token distribution at position $i$, depth $d_i^\pi$. This preserves first-iteration accuracy for easy tokens while training deeper iterations to refine hard tokens.

**Stage 2: Decider imitation under frozen backbone.** We freeze the backbone model $(\theta, \Delta)$ and train the iteration decider $\phi$ to imitate the oracle policy's continuation decisions. We minimize weighted binary cross-entropy:

$$\mathcal{L}_{\text{dec}}(\phi) = -\sum_{i,d} w_d c_i^{(d)} \log \hat{c}_i^{(d)} + (1-c_i^{(d)}) \log(1-\hat{c}_i^{(d)}),$$

where the sum ranges over tokens $i$ and depths $d = 1, \ldots, \min\{d_{\max}-1, d_i^\pi\}$. Here $c_i^{(d)}$ is the ground-truth continuation label, $\hat{c}_i^{(d)}$ is the predicted probability, and $w_d$ is the class weight for label imbalance (ratio of stop to continue labels).

The two-stage scheme stabilizes training by decoupling backbone learning (conditioned on a fixed $\pi$) from iteration policy learning (imitation of $\pi$).

# 6. Experiment

## 6.1. Setup

We present key experiment configurations here, with detailed setups in Appendix A.1.

**Baselines**. We compare diverse methods under equal parameter budgets, using Qwen3-{0.6B,1.7B,4B}-Base (Yang et al., 2025) as backbones. We compare TaH over the following baselines: (1) *Standard*, which always verbalizes directly and reduces to the standard Qwen model; (2) *Soft-Think*, a latent optimization method, implemented following the official design (Zhang et al., 2025b) on top of the Standard model; (3) *AlwaysThink*, similar to TaH, but always iterates twice for all tokens during training and inference; (4) *Ouro*, a looped transformer that can scale iteration depths, implemented following the official design (Zhu et al., 2025). Unless otherwise specified, TaH, Ouro, and AlwaysThink all use a maximum of two iterations, fine-tuned from the same Qwen3 backbone on the same training data and recipe. Ouro reaches its highest performance when set to maximum iterations, so we report results under this setup.

**TaH setup**. To match the total parameter count of TaH (with LoRA and decider) with that of baselines, we prune one layer from the LLM backbone before training. The layer is chosen to minimize the increase in validation loss. We also report results for an unpruned variant, TaH+, which adds less than 3% extra parameters from LoRA and decider. The detailed parameter composition is shown in Table 9. Following (Fu et al., 2025a), we set the continuation threshold $c_{\text{threshold}} = 0.9$ with about 7% of tokens being iterated twice. The oracle policy $\pi$ uses Qwen3-0.6B, 1.7B, and 4B as reference models, respectively.

**Training scheme**. All models are trained on the balanced Open-R1 (Hugging Face, 2025) mixture (math, QA, and code; 100K samples) using supervised fine-tuning. To fit memory and compute limits, we exclude responses longer than 8,192 tokens; 4B models additionally truncate at 4,096 tokens; all other training settings follow the official Open-R1 script. Each method is sufficiently trained for 5 epochs, and we select the checkpoint with the lowest validation loss as the final model. All backbones are initialized from the corresponding Qwen3-Base model.

**Evaluation setup**. We evaluate across challenging reasoning benchmarks, including GSM8K (Cobbe et al., 2021), MATH500 (Hendrycks et al., 2021b), AMC23 (American Mathematics Competitions), AIME25 (American Invitational Mathematics Examination), OlympiadBench (denoted as Olympiad) (He et al., 2024), MBPP++ (Austin et al., 2021), HumanEval++ (denoted as HE++) (Chen et al., 2021), GPQA-Diamond (denoted as GPQA) (Rein et al., 2023), and MMLU-STEM (denoted as MMLU) (Hendrycks et al., 2021a). The maximum generation length is set to 8,192 tokens for all benchmarks, except GSM8K and MMLU-STEM, which use 4,096 due to their simpler problems and larger size. Performance is reported as pass@1 under a zero-shot CoT setting, using a sampling temperature of 0.6. We generate one sample per problem for large datasets (MATH500, OlympiadBench, etc.), and eight samples per problem for small datasets (AMC23, AIME25).

## 6.2. Performance

**Benchmark evaluation**. We validate TaH's reasoning ability across all nine benchmarks. Table 2 presents performance results for models at 0.6B, 1.7B, and 4B parameter sizes. Compared with the strong Standard Qwen3 baselines, we observe that existing approaches (AlwaysThink, SoftThink, and Ouro) yield only marginal improvements when fine-tuned from base. In contrast, TaH achieves consistent gains across benchmarks. For 0.6B and 1.7B models, TaH delivers average improvements of 3.0% and 3.8% over Standard, respectively; TaH+, which adds less than 3% additional parameters, further pushes these gains to 5.3% and 6.2%. Compared to the concurrent work Ouro, TaH and TaH+ achieve 3.8-4.4% and 6.1-6.8% gains, respectively. For 4B models, which are trained with a 4K context length due to resource limits, TaH and TaH+ also achieve gains of 1.7% and 2.2%.

**Hardware-agnostic efficiency**. Tables 19 and 21 report the average iteration depth, per-token FLOPs, and memory access of TaH. On average, TaH performs 1.07 iterations per token. It significantly undercuts the 2.08-2.18× FLOPs and memory access of AlwaysThink, matching the overhead of Standard with only 4-5% additional overhead. See Appendix A.2.5 for more details.

*Table 2.* Accuracy comparison across benchmarks. Best results are highlighted in bold. *4B models are trained with ≤4K lengths due to resource constraints; AlwaysThink is excluded due to Out-Of-Memory (OOM) during training.

| | AIME25 | Olympiad | AMC23 | MATH500 | GSM8K | GPQA | MMLU | HE++ | MBPP++ | Average |
|---|---|---|---|---|---|---|---|---|---|---|
| *0.6B* | | | | | | | | | | |
| Standard | 1.9 | 15.4 | 22.7 | 39.9 | 58.2 | 31.1 | 54.2 | 16.8 | 28.8 | 29.9 |
| SoftThink | 2.9 | 14.0 | 22.2 | 39.6 | 55.9 | 24.7 | 53.0 | 14.3 | 29.5 | 28.5 |
| Ouro | 2.1 | 14.2 | 19.7 | 37.4 | 56.6 | **35.4** | 54.0 | 18.9 | 23.5 | 29.1 |
| AlwaysThink | 1.3 | 12.6 | 21.9 | 37.8 | 52.6 | 30.8 | 51.4 | 9.1 | 13.8 | 25.7 |
| TaH | 2.1 | 19.1 | 24.1 | 46.2 | 63.6 | 29.0 | 56.4 | 21.6 | 33.9 | 32.9/ + 3.0 |
| TaH+ | **4.6** | **20.6** | **24.7** | **51.8** | **67.6** | 31.3 | **59.0** | **22.0** | **35.1** | **35.2**/ + 5.3 |
| *1.7B* | | | | | | | | | | |
| Standard | 10.8 | 33.8 | 39.7 | 67.8 | 80.2 | 30.3 | 74.1 | 39.0 | 51.9 | 47.5 |
| SoftThink | 5.4 | 30.7 | 40.3 | 64.8 | 80.0 | 33.3 | 73.5 | 43.3 | 49.1 | 46.7 |
| Ouro | 10.8 | 31.3 | 40.6 | 68.2 | 79.8 | 32.8 | 72.4 | 40.9 | 45.5 | 46.9 |
| AlwaysThink | 7.5 | 31.0 | 40.9 | 63.2 | 74.2 | 30.5 | 69.6 | 16.4 | 25.6 | 39.9 |
| TaH | 13.8 | 37.2 | 40.9 | 71.4 | **84.8** | 33.3 | 74.8 | 50.0 | 55.3 | 51.3/ + 3.8 |
| TaH+ | **15.4** | **37.6** | **48.4** | **72.6** | 84.5 | **39.4** | **76.6** | **51.5** | **57.5** | **53.7**/ + 6.2 |
| *4B** | | | | | | | | | | |
| Standard | 24.2 | 50.4 | 64.2 | 84.2 | 91.7 | 46.2 | 85.4 | 69.2 | 65.9 | 64.6 |
| SoftThink | 25.8 | 50.2 | 63.1 | 85.0 | **92.5** | **50.3** | 86.0 | 69.2 | 66.7 | 65.4 |
| Ouro | 25.0 | 51.4 | 64.4 | 83.8 | 90.7 | 50.0 | 85.9 | 70.7 | 66.7 | 65.4 |
| TaH | 27.1 | 50.5 | **69.7** | **85.8** | 91.0 | 48.5 | 86.2 | 70.1 | 67.7 | 66.3/ + 1.7 |
| TaH+ | **27.9** | **52.6** | 68.1 | 85.6 | 91.7 | 49.0 | **86.6** | **72.0** | **68.1** | **66.8**/ + 2.2 |

*Table 3.* Real-world decoding performance on a single A800 GPU, tested on AIME25 with 8K max token length.

| | Standard | AlwaysThink | TaH |
|---|---|---|---|
| Avg. Depth | 1.00 | 2.00 | 1.06 |
| Memory (GB) | 4.3 | 6.8 | 4.6 |
| Latency (s) | 210.6 | 747.2 | 301.4 |
| Throughput (token/s) | 38.9 | 11.0 | 27.2 |

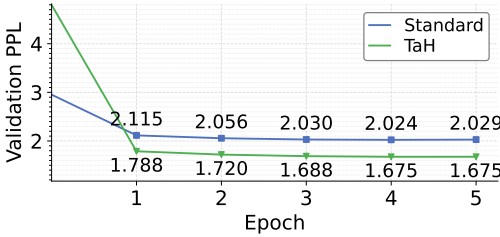

*Figure 4.* Training dynamics of the LLM backbone on Qwen3-0.6B-Base. TaH converges rapidly and achieves lower perplexity.

**Real-world efficiency**. We compare efficiency of 1.7B models at 8K length on a single NVIDIA A800 GPU. As shown in Table 3, TaH iterates twice on only 6% of tokens on AIME25, with 1.48× lower memory overhead and 2.48× faster decoding than AlwaysThink, while achieving higher accuracy. More detailed efficiency experiment setup and runtime breakdown are shown in Appendix A.2.4.

**Training dynamics**. During stage 1 (LLM backbone training), TaH performs iterations according to the oracle policy. As shown in Figure 4, it converges notably faster than the Standard baseline and also achieves much lower validation perplexity. During stage 2 (iteration-decider training), the neural decider successfully imitates the oracle strategy. It reaches about 83% accuracy at predicting the oracle's iteration decisions, as shown in Figure 5.

### 6.3. Design Choice Exploration

We demonstrate the effectiveness and robustness of each design choice of TaH through ablation studies. All experiments in this section train TaH and its variants on the math subset of Open-R1 and evaluate on MATH500, AMC23, and OlympiadBench.

**Model architecture**. (1) **Iteration Scheme**. As shown in Table 4, decider-based iteration outperforms the *Always-1* and *Always-2* alternatives, confirming the practical benefits of selective iteration, even with imperfect decisions from the neural decider. Note that for *Always-1*, duo-causal attention degenerates to regular causal attention. (2) **Duo-Causal Attention**. Replacing duo-causal attention with standard causal attention variants causes significant drops: (a) attend-

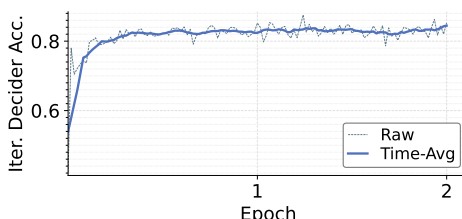

Figure 5. Iteration-decider accuracy vs. epoch (Qwen3-0.6B).

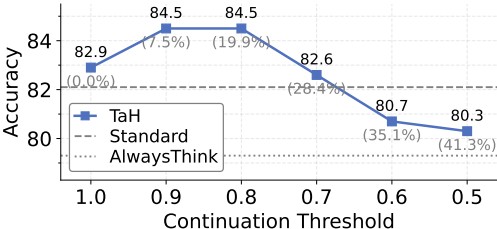

Figure 6. GSM8K accuracy with respect to continuation threshold. Numbers in brackets show the percentage of tokens iterated twice.

ing only to the first iteration (Causal@iter1) drops by 5.4%; (b) attending only to the current iteration (Causal@current) drops even more, by 8.5%. They confirm duo-causal attention's essential role in cross-depth information flow. (3) **Depth Adapters**. Removing LoRA and residual connections leads to consistent drops, confirming their beneficial roles in objective transition across iterations.

**Training scheme**. (1) **Supervision type.** Some previous work supervises all iteration depths with next-token labels, while TaH only supervises final output tokens. As shown in Table 4, such *token+latent* supervision underperforms TaH. This aligns with our intuition that different iterations should focus on their respective objectives. (2) **Iteration policy during LLM training.** We compare our static oracle strategy $\pi$ with two alternatives. The *decider-based* approach trains the iteration decider first, then uses it during backbone training. It suffers from the coupling challenge discussed in Section 5.3. The *dynamic* approach recalculates the oracle using the evolving backbone in Equation 5, facing the same coupling challenge and causing training collapse. These results support our selection of the static oracle policy. (3) **Discrepancy metric in $\pi$.** We compare three discrepancy metrics to trigger latent iteration in $\pi$: top-1 mismatch, entropy, and cross-entropy (detailed definitions in Appendix A.3.2). As Table 4 shows, top-1 mismatch yields the best result, confirming its empirical effectiveness.

**Iteration label robustness**. We further evaluate how sensitive TaH is to the quality of its oracle iteration labels. For this evaluation, all TaH-1.7B variants are trained on the math split while varying only how the iteration depth labels are constructed. As shown in Table 5, labels from a smaller

Table 4. Ablation study on design choices of TaH-0.6B. Each row varies one aspect from the TaH configuration (marked with gray).

| Variant | MATH500 | AMC23 | Olympiad | Average |
|---|---|---|---|---|
| TaH | **51.2** | **32.5** | **23.9** | **35.9** |
| *Model Architecture* | | | | |
| *Iteration Depth* (TaH: Neural decider) | | | | |
| Always-1 | 47.2 | 23.4 | 18.8 | $29.8/{-6.1}$ |
| Always-2 | 32.8 | 15.6 | 10.2 | $19.5/{-16.4}$ |
| *Attention* (TaH: Duo-causal) | | | | |
| Causal@iter1 | 47.8 | 24.4 | 19.4 | $30.5/{-5.4}$ |
| Causal@current | 42.0 | 23.8 | 16.4 | $27.4/{-8.5}$ |
| *Depth Adapters* (TaH: LoRA + Residual) | | | | |
| w/o LoRA | 51.6 | 29.7 | 22.4 | $34.6/{-1.3}$ |
| w/o LoRA & Res. | 49.2 | 22.5 | 21.2 | $31.0/{-4.9}$ |
| *Training Scheme* | | | | |
| *Supervision Type* (TaH: Token-only) | | | | |
| Token+latent | 49.4 | 29.6 | 15.9 | $31.6/{-4.3}$ |
| *Iteration Policy during LLM training* (TaH: Oracle policy) | | | | |
| Decider-based | 44.8 | 24.1 | 17.3 | $28.7/{-7.2}$ |
| Dynamic | 11.0 | 2.8 | 2.7 | $5.5/{-30.4}$ |
| *Discrepancy metric in $\pi$* (TaH: Top1 mismatch) | | | | |
| Cross-entropy | 47.4 | 21.2 | 20.4 | $29.7/{-6.2}$ |
| Entropy | 42.0 | 21.9 | 16.9 | $26.9/{-9.0}$ |

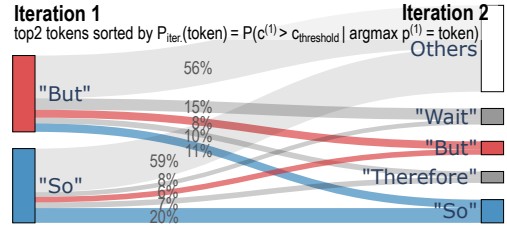

Figure 7. Next-token prediction changes across iterations. Top-2 frequent iterated tokens are visualized.

0.6B reference and labels with 10% random flips both still improve over Standard-1.7B. The clean same-scale 1.7B reference gives the strongest average gain, so we use this label source in the main experiments.

**Continuation threshold**. TaH shows robust performance across continuation thresholds and iteration ratios (Figure 6). We empirically set $c_{\text{threshold}} = 0.9$ for all evaluations.

**Deeper iteration depth**. We train 1.7B variants with larger maximum iteration depths on the math subset. TaH-3 and TaH-4 set $D_{\max} = 3$ and $D_{\max} = 4$, respectively, following the same sparse, decreasing allocation principle as TaH-2: most tokens stop at depth 1, while harder tokens continue through additional latent passes. As shown in Table 6, the average gain over Standard increases from +5.0% (TaH-2) to +6.0% (TaH-3) and +6.2% (TaH-4) with little extra cost.

*Table 5.* Robustness of TaH-1.7B to imperfect oracle iteration labels. Each robustness row varies the label construction from the main TaH configuration (marked with gray); "10% noise" randomly flips labels generated by the Qwen3-1.7B reference.

| Method | Iter. Label | MATH500 | GSM8K | AMC23 | Olympiad | AIME25 | Average |
|---|---|---|---|---|---|---|---|
| *Primary baselines* | | | | | | | |
| Standard-1.7B | – | 68.4 | 82.1 | 42.2 | 33.0 | 13.3 | 47.8 |
| TaH-1.7B | 1.7B reference | **74.4** | **84.5** | **48.4** | **38.8** | **17.9** | **52.8**/+5.0 |
| *Label robustness checks* | | | | | | | |
| TaH-1.7B | 0.6B reference | 70.2 | 82.3 | 43.4 | 33.5 | 14.2 | 48.7/+0.9 |
| TaH-1.7B | 1.7B reference + 10% noise | 73.6 | 83.5 | 46.8 | 36.0 | 14.6 | 50.9/+3.1 |

*Table 6.* Performance comparison with larger maximum iteration depths. TaH variants are all 1.7B, trained on the math subset. Brackets show the percentage of tokens executing iterations 2, 3, and 4.

| Method | Iter. Policy | MATH500 | GSM8K | AMC23 | Olympiad | AIME25 | Avg. |
|---|---|---|---|---|---|---|---|
| Standard | – | 68.4 [0.0, 0.0, 0.0] | 82.1 [0.0, 0.0, 0.0] | 42.2 [0.0, 0.0, 0.0] | 33.0 [0.0, 0.0, 0.0] | 13.3 [0.0, 0.0, 0.0] | 47.8 [0.0, 0.0, 0.0] |
| TaH-2 | | 74.4 [5.6, 0.0, 0.0] | 84.5 [7.5, 0.0, 0.0] | 48.4 [4.2, 0.0, 0.0] | 38.8 [5.7, 0.0, 0.0] | 17.9 [6.0, 0.0, 0.0] | 52.8 [5.8, 0.0, 0.0] |
| TaH-3 | Decreasing | 74.8 [6.5, 1.2, 0.0] | 84.0 [8.6, 1.8, 0.0] | 49.1 [6.3, 1.0, 0.0] | 41.6 [5.3, 1.0, 0.0] | 19.6 [5.4, 1.0, 0.0] | 53.8 [6.4, 1.2, 0.0] |
| TaH-4 | | 74.8 [4.9, 0.6, 0.6] | 84.6 [7.1, 0.9, 0.8] | 49.7 [4.3, 0.5, 0.4] | 40.7 [4.0, 0.5, 0.4] | 20.4 [4.3, 0.5, 0.3] | 54.0 [4.9, 0.6, 0.5] |
| TaH-3 | Uniform | 70.2 [29.4, 31.8, 0.0] | 82.5 [29.5, 35.8, 0.0] | 41.6 [30.7, 32.9, 0.0] | 34.4 [30.6, 34.1, 0.0] | 14.2 [30.4, 35.3, 0.0] | 48.6 [30.1, 34.0, 0.0] |

*Table 7.* Performance of 1.7B models under restricted inference depth, grouped by task category.

| Method | Train. Depth | Infer. Depth | Math | Science | Code | Avg. |
|---|---|---|---|---|---|---|
| Standard | 1 | 1 | 46.5 | 52.2 | 45.5 | 47.5 |
| TaH | 2 | 2 | 49.6 | 54.1 | 52.7 | 51.3/+3.8 |
| | 2 | 1 | 48.1 | 53.9 | 49.2 | 49.6/+2.1 |
| TaH+ | 2 | 2 | **51.7** | **58.0** | **54.5** | **53.7**/+6.2 |
| | 2 | 1 | 50.2 | 52.4 | 49.8 | 50.6/+3.1 |

*Figure 8.* Distribution of token prediction accuracy across iterations for Ouro and TaH.

We further study the impact of the label distribution by comparing with TaH-3-Uniform, which assigns same number of tokens across depths. The weaker result of the uniform variant confirms that sparser allocation of deeper iterations is more effective than forcing frequent use of deeper depths. Detailed setup is provided in Appendix A.1.3.

**Restricted inference depth**. We evaluate whether the benefits of TaH persist when the maximum inference depth is below the training maximum. Table 7 evaluates 1.7B models trained with $D_{max} = 2$ but forced to verbalize after the first iteration at inference. Although weaker than full TaH inference, restricted-depth inference still outperforms Standard by 2.1-3.1% on average, suggesting that TaH training also improves first-iteration representations.

### 6.4. Behavior Analysis

**Token alternation patterns**. We analyze which tokens TaH selects for deeper iteration. As shown in Figure 7, *But* and *So* are iterated most frequently. These tokens are hard to predict because they mark points of contrast or causality that

can redirect subsequent reasoning. At such junctures, TaH uses additional iterations to refine its reasoning direction. See Appendix A.3.4 for details.

**Attention pattern**. We visualize the attention pattern of TaH in Figure 14 and Appendix A.3.5. Duo-causal attention focuses on different iterations in different heads, extracting broader contexts from multiple depths.

**Accuracy landscape**. Figure 8 shows NTP accuracy across iterations. Because Ouro trains all iterations to predict all tokens, predictable tokens across depths largely overlap. TaH specializes deeper iterations for hard tokens, improving overall coverage under selective iteration.

## 7. Conclusion

We present TaH, a selective latent iteration method that simultaneously improves reasoning performance and efficiency. TaH introduces duo-causal attention, depth-specific LoRA, and a neural iteration decider, stably trained in two stages under a static oracle policy. Across nine benchmarks, TaH improves accuracy by 5.3-6.2% over strong baselines with minimal overhead, opening a new direction for better reasoning under parameter constraints.

## Acknowledgements

This work was supported by National Natural Science Foundation of China (No. 62506197, 62325405, 62104128, U19B2019, U21B2031, 61832007, 62204164, 92364201), Tsinghua EE Xilinx AI Research Fund, and Beijing National Research Center for Information Science and Technology (BNRist). We thank Xuefei Ning, and Donglin Yang for their valuable discussions and suggestions. We also thank all the support from Infinigence-AI.

## Impact Statement

This paper presents work whose goal is to advance the field of Looped Transformers. There are many potential societal consequences of our work, none of which we feel must be specifically highlighted here.

From a practical perspective, this paper improves model performance for parameter-constrained language models through selective iteration. While this is an early exploration in this direction, we observe significant potential to simultaneously achieve accuracy gains comparable to much larger models while skipping iterations on most tokens. By enhancing smaller models, our work may help democratize access to capable AI systems across diverse settings.

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

# A. Appendix

## A.1. Additional Experiment Setups

### A.1.1. TRAINING RECIPE

We follow the official training setup of Open-R1 (Hugging Face, 2025) and use the Mixture-of-Thoughts dataset. For our main performance experiments (Section 6.2), we filter samples with output length exceeding 8K tokens from each category (math, code, and QA), then randomly sample 33K examples from each to form a balanced 100K training set. The filtered dataset contains 480M tokens, with 1% reserved for validation. For design choice exploration (Section 6.3), we use only the math subset filtered by 8K output length, resulting in 75K training samples. For 0.6B and 1.7B models, we use a maximum sequence length of 8192 tokens for all methods. For 4B models, we reduce the maximum sequence length to 4096 for Standard, TaH, TaH+, and 3072 for Ouro due to memory constraints. Detailed training hyperparameters are listed in Table 8.

_Table 8._ Training hyperparameters.

| Hyperparameter | Value |
| --- | --- |
| learning rate | 4e-5 |
| max grad norm | 0.2 |
| training epochs | 5 |
| global batch size | 128 |
| warmup ratio | 0.03 |
| lr scheduler | cosine (min-lr ratio 0.1) |
| precision | bfloat16 |

### A.1.2. BASELINE SETUPS

**AlwaysThink**. AlwaysThink uses the same architecture as TaH, except that it iterates twice at every token position and uses standard causal attention (attending only to the current iteration depth) instead of duo-causal attention.

**Ouro**. We implement the Ouro architecture following its official design (Zhu et al., 2025), but fine-tune it on the Open-R1 dataset from Qwen-Base initialization, to align with all other methods. During fine-tuning, we adopt the entropy-regularized loss of Ouro. Since the original Ouro is trained from scratch on different data distributions, performance may differ from the original paper.

### A.1.3. ITERATION DEPTH FLEXIBILITY

The main experiments in this paper focus on $D_{\max} = 2$. Here we detail how the oracle iteration policy generalizes to arbitrary maximum iteration depth, as used in Section 6.3. The focus of this work is how to *selectively attribute* iterations to different tokens, rather than increasing the average iteration depth.

**Binary case** ($D_{\max} = 2$). Prior work has proposed many ways to estimate token difficulty, including excess loss (Lin et al., 2024; Xie et al., 2023), entropy (Wang et al., 2025b; Chen et al., 2023), and prediction difference (Fu et al., 2025a). For shallow iteration budgets of up to two, we adopt the prediction-difference policy used in the main experiments: tokens that fail top-1 next-token prediction at the first iteration are labeled as hard. Formally, the oracle iteration depth $d_i^\pi$ follows a binary rule:

$$d_i^\pi = \begin{cases} 1, & \text{if } h_i = 0 \quad \text{(easy token)} \\ D_{\max}, & \text{if } h_i = 1 \quad \text{(hard token)}, \end{cases} \tag{8}$$

where $h_i$ is the hard-token indicator. This induces a Bernoulli distribution over depths: easy tokens verbalize at depth 1, hard tokens iterate to depth $D_{\max}$.

**General framework** ($D_{\max} > 2$). For deeper iteration budgets, we replace the binary rule with a continuous difficulty score and a target depth distribution. We use the reference model's first-pass cross-entropy as a continuous measure of token difficulty:

$$\ell_i^{\text{ref}} = -\log p_{i,\text{ref}}^{(1)}(t_{i+1}).$$

Let $\Lambda$ be a target distribution over $\{1, \ldots, D_{\max}\}$ specifying the desired fraction of tokens at each depth. We assign iteration depths via quantile mapping: rank all tokens by $\ell^{\text{ref}}$ to obtain $u_i = \text{QuantileRank}(\ell_i^{\text{ref}}) \in [0, 1]$, then set

$$d_i^\pi = F_\Lambda^{-1}(u_i), \tag{9}$$

where $F_\Lambda^{-1}$ is the quantile function (inverse CDF) of $\Lambda$. Harder tokens (higher loss) thus receive deeper iterations, and the overall depth distribution follows $\Lambda$. The binary case is recovered when $\Lambda$ matches the depth distribution induced by the

hard-token labels (i.e., mass at depths 1 and $D_{\max}$ equal to the easy and hard fractions, respectively). The oracle depth $d_i^\pi$ is then converted into per-depth supervision for the iteration decider (Section 5.3): at each depth $d \in \{1, \ldots, D_{\max} - 1\}$, the continuation label $c_i^{(d)} = \Bbbk[d < d_i^\pi]$ equals 1 (continue iterating) when the token has not yet reached its assigned depth, and 0 (verbalize) once $d = d_i^\pi$. At inference, the decider's predicted continuation probabilities $\hat{c}_i^{(d)}$ are compared against the threshold $c_{\text{threshold}}$ to decide whether each token continues at each depth.

**Deeper-depth experiment setup**. For TaH-3-Decreasing in Table 6, we set the training target distribution over depths to $\Lambda = (0.88, 0.06, 0.06)$. For TaH-4-Decreasing, we set $\Lambda = (0.88, 0.04, 0.04, 0.04)$. That is, we keep the depth-1 fraction the same as TaH-2 and evenly allocate the remaining budget to deeper depths; the inference continuation threshold is set to $c_{\text{threshold}} = 0.9$, which makes the realized continuation rate decrease with depth. For comparison, TaH-3-Uniform uses $\Lambda = (0.34, 0.33, 0.33)$ during training and $c_{\text{threshold}} = 0.5$ during inference to preserve a roughly uniform depth allocation. The realized continuation ratios are reported in brackets in Table 6.

### A.1.4. PARAMETER BREAKDOWN

Table 9 reports the parameter breakdown of the Standard, TaH, and TaH+ methods. To offset the additional parameters introduced by TaH through LoRA and the iteration decider, we remove one layer from the LLM backbone, ensuring a fair comparison. In practical deployments, we recommend TaH+, which adds only about 3% additional parameters.

## A.2. Additional Experimental Results

### A.2.1. MATH-ONLY TRAINING

This section isolates the math-only setting: models are trained on the math subset of Open-R1 and evaluated on math benchmarks as well as out-of-domain tasks.

**Math benchmark evaluation**. In this section, all models are trained on the math subset of Open-R1 and evaluated on math benchmarks (see Section A.1.1 for training details). We also add a *Routing* baseline, which selects a model from a candidate pair for each question. In our experiments, we use two pairs: (1) MobileLLM-R1-360M, Qwen3-1.7B, and (2) Qwen3-0.6B, Qwen3-4B. All candidate models are SFT-trained under the same settings. For each pair, the routing ratio is calibrated so that the average active parameter count matches our 0.6B and 1.7B targets, respectively. Table 10 reports results for 0.6B, 1.7B, and 4B backbones across five challenging math benchmarks. Even with strong Qwen3-Base initialization, existing approaches show limited effectiveness: AlwaysThink and routing methods fail to consistently outperform the standard baseline, while SoftThink yields only marginal gains. In contrast, TaH achieves stable improvements, with average gains of 4.0% (0.6B), 5.0% (1.7B), and 3.8% (4B) over Standard. TaH+, which adds less than 3% more parameters, further improves to 5.3%, 5.4%, and 4.2%, respectively. For 0.6B and 1.7B, TaH achieves 8.1-11.3% gains over AlwaysThink, and TaH+ achieves 8.5-12.6% gains. AlwaysThink-4B is not evaluated due to out-of-memory during training.

**Out-of-domain evaluation**. We further evaluated the zero-shot generalization capability of models trained solely on math datasets from the main paper. As shown in Table 11, TaH+ demonstrates consistent improvements not only on in-domain math benchmarks (MATH500, AMC23) but also on out-of-domain tasks like MMLU-STEM. This indicates that the thinking

*Table 9.* Parameter breakdown of Standard, TaH, and TaH+. Counts are reported using M (million) and B (billion).

| Param. | Method | Backbone | LoRA | Iter. Decider | Total |
|---|---|---|---|---|---|
| *0.6B* | Standard | 596M | – | – | 596M |
| | TaH | 580M | 10M | 5M | 595M |
| | TaH+ | 596M | 10M | 5M | 611M |
| *1.7B* | Standard | 1.72B | – | – | 1.72B |
| | TaH | 1.67B | 17M | 18M | 1.71B |
| | TaH+ | 1.72B | 17M | 18M | 1.76B |
| *4B* | Standard | 4.02B | – | – | 4.02B |
| | TaH | 3.92B | 32M | 70M | 4.02B |
| | TaH+ | 4.02B | 33M | 70M | 4.12B |

*Table 10.* Accuracy comparison of different baselines across five benchmarks and three model sizes. Subscripts indicate improvement over Standard. The top two scores for each task and model size are highlighted in bold.

| Param. | Benchmark | Method | | | | | |
|---|---|---|---|---|---|---|---|
| | | Standard | Routing | SoftThink | AlwaysThink | TaH | TaH+ |
| 0.6B | AIME25 | **4.2** | 1.0 | 2.5 | 1.5 | **4.2** | **5.0** |
| | OlympiadBench | 18.8 | 7.4 | 19.4 | 10.2 | **23.9** | **24.0** |
| | AMC23 | 23.4 | 10.9 | 24.1 | 15.6 | **32.5** | **30.6** |
| | MATH500 | 47.2 | 27.3 | 48.8 | 32.8 | **51.2** | **54.2** |
| | GSM8K | 62.5 | 45.6 | 61.3 | 54.6 | **64.4** | **68.8** |
| | Average | 31.2 | 18.5 | 31.2 | 22.9 | **35.2**$_{/+4.0}$ | **36.5**$_{/+5.3}$ |
| 1.7B | AIME25 | 13.3 | 10.2 | 12.9 | 10.0 | **17.9** | **14.6** |
| | OlympiadBench | 33.0 | 30.6 | 33.4 | 30.0 | **38.8** | **41.2** |
| | AMC23 | 42.2 | 42.2 | 43.1 | 42.5 | **48.4** | **51.2** |
| | MATH500 | 68.4 | 60.0 | 68.8 | 61.8 | **74.4** | **73.0** |
| | GSM8K | 82.1 | 71.2 | 79.6 | 79.3 | **84.5** | **85.8** |
| | Average | 47.8 | 36.8 | 47.6 | 44.7 | **52.8**$_{/+5.0}$ | **53.2**$_{/+5.4}$ |
| 4B | AIME25 | 23.3 | 22.5 | 22.5 | | **30.4** | **28.3** |
| | OlympiadBench | 47.7 | 45.0 | 50.1 | | **50.5** | **52.0** |
| | AMC23 | 62.8 | 60.9 | 64.1 | OOM | **70.3** | **70.6** |
| | MATH500 | 82.8 | 76.1 | 83.2 | | **84.4** | **85.6** |
| | GSM8K | 90.5 | 85.3 | **90.9** | | 90.4 | **91.5** |
| | Average | 61.4 | 58.0 | 62.2 | – | **65.2**$_{/+3.8}$ | **65.6**$_{/+4.2}$ |

patterns learned by TaH+ on math problems are robust and transferrable to broader scientific reasoning tasks.

### A.2.2. ADDITIONAL ORACLE ANALYSIS

We provide two complementary oracle analyses. (1) **Downstream generation.** Table 12 reports generation results on MATH100 with Qwen3-0.6B. We use DeepSeek-R1-Distill-Qwen-32B predictions (Guo et al., 2025) as proxy labels. The oracle verbalizes when the model's top-1 prediction matches the proxy label, and iterates otherwise. With our trained iteration decider approximating the oracle, TaH outperforms both Standard and AlwaysThink baselines. However, the ideal oracle policy achieves even higher gains, indicating future potential. (2) **Next-token prediction.** Figure 9 shows prediction transitions obtained by verbalizing tokens from all iteration depths. It reveals that AlwaysThink produces more incorrect than correct revisions, demonstrating latent overthinking. In contrast, oracle-controlled iterations substantially increase correct revisions by selectively targeting hard tokens.

*Table 11.* Performance of math-only trained models (0.6B and 1.7B) on in-domain math benchmarks and the out-of-domain STEM benchmark (MMLU-STEM).

| Param. | Benchmark | Standard | SoftThink | AlwaysThink | TaH+ |
|---|---|---|---|---|---|
| 0.6 | MATH500 | 47.2 | 48.8 | 32.8 | **54.2** |
| | AMC23 | 23.4 | 24.1 | 15.6 | **30.6** |
| | MMLU-STEM | 51.6 | 51.4 | 42.6 | **56.3** |
| | Average | 40.7 | 41.4 | 30.3 | **47.0** |
| 1.7 | MATH500 | 68.4 | 68.8 | 61.8 | **73.0** |
| | AMC23 | 42.2 | 43.1 | 42.5 | **51.2** |
| | MMLU-STEM | 70.8 | 70.6 | 63.8 | **73.7** |
| | Average | 60.5 | 60.8 | 56.0 | **66.0** |

*Table 13.* Comparison with a Standard-Pruned baseline on 1.7B math-trained models. Standard-Pruned removes the same backbone layer as TaH but does not use latent iteration.

| Method | AIME25 | Olympiad | AMC23 | MATH500 | GSM8K | Average |
|---|---|---|---|---|---|---|
| Standard | 13.3 | 33.0 | 42.2 | 68.4 | 82.1 | 47.8 |
| Standard-Pruned | 11.7 | 32.7 | 41.3 | 68.0 | 79.4 | 46.6 |
| TaH | **17.9** | **38.8** | **48.4** | **74.4** | **84.5** | **52.8** |

*Table 14.* Performance on MATH500 and GSM8K-500 (first 500 GSM8K samples)

| Dataset | Method | |
|---|---|---|
| | Standard-0.6B | Ponder-1.4B |
| MATH500 | 47.2 | 2.0 |
| GSM8K-500 | 62.8 | 1.8 |
| Avg. | 55 | 1.9 |

*Table 12.* MATH100 accuracy under different training and inference policies on Qwen3-0.6B, using DeepSeek-R1-Distill-Qwen-32B predictions as proxy labels.

| Training | Inference | Accuracy |
|---|---|---|
| Standard | Standard | 52 |
| AlwaysThink | AlwaysThink | 38/−14 |
| AlwaysThink | TaH-Oracle | 77/+25 |
| TaH-Oracle | TaH-Decider | 54/+2 |
| TaH-Oracle | TaH-Oracle | 80/+28 |

*Figure 9.* Next-token prediction transitions across iteration depths on Qwen3-0.6B.

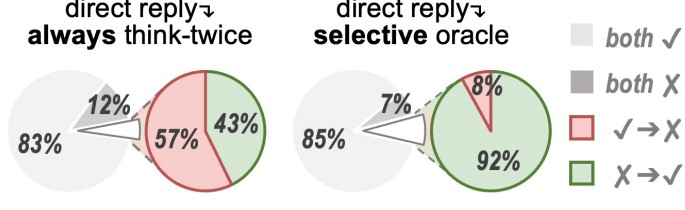

### A.2.3. ADDITIONAL BASELINES

**Standard-pruned baseline**. Since TaH removes one backbone layer to match the parameter budget of Standard, we additionally compare against a Standard-Pruned baseline with the same layer removed but without selective iteration. Table 13 shows that pruning alone slightly degrades performance, while TaH substantially outperforms both Standard and Standard-Pruned under the same training and evaluation setup. This isolates the gain from selective latent iteration rather than from pruning.

**Additional latent thinking baselines**. Some latent thinking methods require pre-training and use base models other than Qwen3. We also compare with these methods, including Ponder (Zeng et al., 2025). Specifically, we adopt the released pretrained PonderingPythia-1.4B as the base model and perform SFT on the same training data. We observe that the fine-tuned model learns the stylistic patterns of the training data, but still underperforms substantially, which may be attributable to the limited capability of the PonderingPythia-1.4B backbone.

### A.2.4. REAL-WORLD EFFICIENCY

**Setup**. We investigate the real-world efficiency of different 1.7B models under our current implementation. All measurements were obtained on a single A800 GPU with a batch size of 1 and a maximum output length of 8192 tokens, using a challenging AIME25 problem where all three methods reached the token limit. Memory usage was profiled using `torch.cuda.memory._record_memory_history`.

**Latency breakdown**. We report the decoding latency, throughput, and a detailed time breakdown for Standard, AlwaysThink, and TaH in Table 15. Here, *Iter-1 forward* and *Iter-2 forward* denote the total forward-pass time spent on the first and second latent iterations, respectively; *Iter decider* is the time for the iteration decider network to judge whether to continue iterating or verbalize; *LoRA switching* is the overhead of switching LoRA adapters; and *Other* includes tensor initialization,

*Table 15.* Per-component latency breakdown on a single A800 GPU.

| Component | Standard | | TaH | | AlwaysThink | |
|---|---|---|---|---|---|---|
| | Latency (s) | Ratio(%) | Latency (s) | Ratio(%) | Latency (s) | Ratio(%) |
| Iter-1 Forward | 210.6 | 100.0 | 229.8 | 76.2 | 224.1 | 30.0 |
| Iter-2 Forward | – | – | 29.6 | 9.8 | 384.7 | 51.5 |
| Iter. Decider | – | – | 10.5 | 3.5 | – | – |
| LoRA Switching | – | – | 7.5 | 2.5 | 91.1 | 12.2 |
| Other | – | – | 24.1 | 8.0 | 47.4 | 6.3 |

concatenation, and related bookkeeping.

**Discussion**. We note that our current implementation is not yet optimized at the system level, so there remains room for further efficiency improvements. For example, the *LoRA Switching* and *Other* overheads (bookkeeping) are relatively high due to the Python-level implementation of dynamic control flow. These engineering optimizations are largely orthogonal to the algorithmic design of TaH, and we plan to continue refining the implementation to further reduce latency and memory overhead. The theoretical FLOPs and memory access analysis of TaH are provided in Appendix A.2.5.

A.2.5. THEORETICAL EFFICIENCY ANALYSIS

Following prior work (Hoffmann et al., 2022; Yang et al., 2024; Ma et al., 2025), we analyze the per-token decoding computation and memory access via analytical operator-level profiling: we trace the forward pass symbolically, accumulating FLOPs and memory bytes from tensor shapes and data types across all modules.

**Notation and formulation**. Table 16 lists the symbols used throughout this analysis; Table 17 breaks down the cost from individual operators (`nn.linear`, `nn.sdpa` for `scaled_dot_product_attention`) through per-layer modules to a full decode step. Since $h = n_h h_h$ for our LLMs, some terms are simplified accordingly. Lightweight vector ops (RMSNorm, RoPE, SiLU, residual add) are also exhaustively profiled and included in the final numbers in Table 21. Since they generally contribute very little to the total cost, they are omitted from the formulas below for brevity.

**Standard decoding**. For a standard LLM generating $t_{\text{out}}$ output tokens from a prefill of $t_{\text{in}}$ input tokens, the total FLOPs and memory access are:

$$\sum_{i=1}^{t_{\text{out}}} \text{FLOPs}_{\text{step}}(t_{\text{in}}+i-1), \qquad \sum_{i=1}^{t_{\text{out}}} \text{Mem}_{\text{step}}(t_{\text{in}}+i-1). \tag{10}$$

Per-token averages are obtained by dividing by $t_{\text{out}}$.

*Table 16.* Notation for theoretical efficiency analysis.

| Symbol | Description | Symbol | Description |
|---|---|---|---|
| $h$ | Hidden dim | $n_h$ | Query head count |
| $s$ | KV cache length | $n_{\text{kv}}$ | KV head count |
| $d$ | Iteration depth | $h_h$ | Per-head dim ($h/n_h$) |
| $l$ | Number of layers | $h_{\text{kv}}$ | Total KV dim ($n_{\text{kv}}h_h$) |
| $w$ | Weight count per layer | $h_{\text{ff}}$ | FFN intermediate size |
| $b$ | Bytes per element | $t_{\text{in}}$ | Input token count |
| $v$ | Vocabulary size | $t_{\text{out}}$ | Output token count |
| $k$ | Top-$k$ logits | | |

**Extension to TaH**. TaH assigns each of the $t_{\text{out}}$ output tokens ($i = 1, \ldots, t_{\text{out}}$) a depth $d_i \in \{1, \ldots, d_{\max}\}$. Generating one output token at depth $d_i$ requires $d_i$ backbone passes, $d_i-1$ inner transitions (Equation 4), and $\min(d_i, d_{\max}-1)$ decider calls (one after each backbone pass; the call at $d_{\max}$ is skipped because the token must verbalize). The total decoding FLOPs

*Table 17.* FLOPs and memory access per decode token, from operators to a full decode step. The Qwen architecture uses GQA ($g=n_h/n_{kv}$) and gated FFN (SwiGLU). Self-Attention applies 4 nn.linear (Q,K,V,O) plus nn.sdpa; FFN applies 3 nn.linear (gate, up, down). $w=2h(h+h_{kv})+3hh_{ff}$. Minor terms are absorbed into $\mathcal{O}(\cdot)$.

| Component | FLOPs | Memory (bytes) |
|---|---|---|
| *Operators* | | |
| nn.linear | $2\,h_{in}\,h_{out}$ | $(h_{in}h_{out}+h_{in}+h_{out})\,b$ |
| nn.sdpa | $4\,s\,h$ | $[2h+s(2h_{kv}+3n_h)]\,b$ |
| *Modules (per layer)* | | |
| FFN | $6\,h\,h_{ff}$ | $3\,h\,h_{ff}\,b + \mathcal{O}(hb)$ |
| Self-Attention | $4h(h+h_{kv})+4sh$ | $[2h(h+h_{kv})+s(2h_{kv}+3n_h)]\,b+\mathcal{O}(hb)$ |
| *Decode step ($\times l$ layers, at length $s$)* | | |
| Step | $l[4h(h+h_{kv})+4sh+6hh_{ff}]+\mathcal{O}(lh_{ff})$ | $l[w+s(2h_{kv}+3n_h)]\,b+\mathcal{O}(lhb)$ |

and memory access over $t_{out}$ output tokens are:

$$\sum_{i=1}^{t_{out}} \Big[\ \sum_{d=1}^{d_i} \text{FLOPs}_{step}\big(s_i^{(d)}\big) + (d_i-1)\,\text{FLOPs}_{it} + \min(d_i, d_{max}-1)\,\text{FLOPs}_{dec}\ \Big], \qquad (11)$$

$$\sum_{i=1}^{t_{out}} \Big[\ \sum_{d=1}^{d_i} \text{Mem}_{step}\big(s_i^{(d)}\big) + (d_i-1)\,\text{Mem}_{it} + \min(d_i, d_{max}-1)\,\text{Mem}_{dec}\ \Big], \qquad (12)$$

where $s_i^{(d)}$ is the KV cache length visible at depth $d$ for the $i$-th output token under duo-causal attention (Equation 6). Since the KV cache at each depth is visible to all deeper levels, for $d_{max}=2$ we have $s_i^{(1)} = t_{in} + i - 1$ (all prior tokens pass through depth 1) and $s_i^{(2)} = s_i^{(1)} + |\{j < i : d_j=2\}|$ (depth-1 entries plus prior depth-2 tokens). Table 18 lists the per-call cost of the inner transition and iteration decider; both are negligible relative to one backbone pass. The overhead is therefore dominated by the average depth $\bar{d} = \frac{1}{t_{out}} \sum_i d_i$, giving approximately $\bar{d}\times$ the standard cost when $\bar{d}$ is small.

**Extension to AlwaysThink**. For AlwaysThink, every output token iterates to the maximum depth. When $d_i=d_{max}=2$ for all $i$, every depth-1 KV entry from output tokens has a corresponding depth-2 entry, giving $s_i^{(2)} \approx 2\,s_i^{(1)}$. The two backbone passes incur $2\times$ the single-pass FFN cost, while their combined attention cost scales as $\mathcal{O}(s_i^{(1)}) + \mathcal{O}(s_i^{(2)}) \approx 3\,\mathcal{O}(s_i^{(1)})$, i.e., $\sim3\times$ the single-pass attention FLOPs. Combining both components yields 2-3$\times$ total overhead over Standard.

*Table 18.* Per-call cost of TaH-specific components. The inner transition is applied $d_i-1$ times and the decider $\min(d_i, d_{max}-1)$ times per token. Both are negligible relative to one backbone pass.

| Component | Description | FLOPs | Memory (bytes) |
|---|---|---|---|
| Inner transition | $x_i^{(d+1)}=p_i^{(d)} E$ (Eq. 4) | $v + 2hk$ | $(v+kh+h)\,b$ |
| Iter. decider $\mathcal{I}_\phi$ | MLP ($|\phi|$ params) | $2|\phi|$ | $(|\phi|+\text{acts})\,b$ |

**Results**. Tables 19 and 20 report the average number of input/output tokens and latent iterations per token across benchmarks. We plug these statistics into the analysis to obtain the per-token costs in Table 21. With only a small fraction of tokens iterating twice, TaH incurs only 1.04-1.05$\times$ the cost of the Standard baseline. In contrast, AlwaysThink sets $\bar{d}=2$, requiring 2.19-2.27$\times$ more computation and memory access. These results confirm that TaH achieves the performance benefits while avoiding the substantial efficiency penalty.

## A.3. Additional Analysis

### A.3.1. TRAINING SCHEME DYNAMICS

Figure 10 complements the training-scheme ablations in Table 4 with validation-perplexity curves. Token-only supervision with a fixed oracle policy trains stably, while token+latent supervision and decoder-based training lag behind. The dynamic oracle can reduce validation perplexity but is unstable during generation, matching its poor downstream accuracy. For the

*Table 19.* Input tokens (shared across methods) and output token / iteration statistics for Standard, AlwaysThink, TaH, and TaH+ (General-trained version).

| Param. | Dataset | In. | Standard | | AlwaysThink | | Ouro | | TaH | | TaH+ | |
|--------|---------|-----|------|------|------|------|------|------|------|------|------|------|
| | | | Out. | Iter. | Out. | Iter. | Out. | Iter. | Out. | Iter. | Out. | Iter. |
| | AIME25 | 159 | 7687 | 1.00 | 6885 | 2.00 | 7370 | 2.00 | 7741 | 1.04 | 7677 | 1.05 |
| | OlympiadBench | 100 | 6937 | 1.00 | 6175 | 2.00 | 6733 | 2.00 | 6926 | 1.04 | 6796 | 1.04 |
| | AMC23 | 85 | 6602 | 1.00 | 6332 | 2.00 | 6355 | 2.00 | 6626 | 1.04 | 6688 | 1.04 |
| | MATH500 | 71 | 5364 | 1.00 | 5072 | 2.00 | 5274 | 2.00 | 5189 | 1.04 | 5092 | 1.04 |
| 0.6B | GSM8K | 61 | 2144 | 1.00 | 1710 | 2.00 | 2039 | 2.00 | 1895 | 1.06 | 1808 | 1.06 |
| | GPQA | 234 | 6533 | 1.00 | 5998 | 2.00 | 6087 | 2.00 | 6104 | 1.11 | 5966 | 1.12 |
| | MMLU-STEM | 97 | 2302 | 1.00 | 2134 | 2.00 | 2201 | 2.00 | 2057 | 1.14 | 2033 | 1.14 |
| | HumanEval++ | 133 | 4711 | 1.00 | 4411 | 2.00 | 4149 | 2.00 | 4045 | 1.05 | 3621 | 1.05 |
| | MBPP++ | 51 | 4652 | 1.00 | 4964 | 2.00 | 4521 | 2.00 | 3094 | 1.05 | 3175 | 1.05 |
| | Average ratio | – | $1.00\times$ | $1.00\times$ | $0.93\times$ | $2.00\times$ | $0.95\times$ | $2.00\times$ | $0.93\times$ | $1.06\times$ | $0.91\times$ | $1.07\times$ |
| | AIME25 | 159 | 7436 | 1.00 | 7458 | 2.00 | 7621 | 2.00 | 7799 | 1.06 | 7547 | 1.07 |
| | OlympiadBench | 100 | 6190 | 1.00 | 6318 | 2.00 | 6505 | 2.00 | 6555 | 1.06 | 6403 | 1.07 |
| | AMC23 | 85 | 5918 | 1.00 | 5766 | 2.00 | 6261 | 2.00 | 6423 | 1.06 | 5989 | 1.07 |
| | MATH500 | 71 | 4159 | 1.00 | 4347 | 2.00 | 4455 | 2.00 | 4474 | 1.06 | 4076 | 1.07 |
| 1.7B | GSM8K | 61 | 1540 | 1.00 | 1775 | 2.00 | 1745 | 2.00 | 1718 | 1.07 | 1597 | 1.09 |
| | GPQA | 234 | 6269 | 1.00 | 6400 | 2.00 | 6380 | 2.00 | 6729 | 1.16 | 6502 | 1.18 |
| | MMLU-STEM | 97 | 1774 | 1.00 | 1729 | 2.00 | 2005 | 2.00 | 1893 | 1.14 | 1769 | 1.18 |
| | HumanEval++ | 133 | 3875 | 1.00 | 3826 | 2.00 | 4177 | 2.00 | 3905 | 1.07 | 3797 | 1.09 |
| | MBPP++ | 51 | 3438 | 1.00 | 3834 | 2.00 | 3682 | 2.00 | 3181 | 1.07 | 3035 | 1.09 |
| | Average ratio | – | $1.00\times$ | $1.00\times$ | $1.02\times$ | $2.00\times$ | $1.05\times$ | $2.00\times$ | $1.05\times$ | $1.08\times$ | $1.00\times$ | $1.10\times$ |
| | AIME25 | 159 | 7007 | 1.00 | – | – | 7159 | 2.00 | 7452 | 1.04 | 7282 | 1.04 |
| | OlympiadBench | 100 | 5699 | 1.00 | – | – | 5649 | 2.00 | 6010 | 1.05 | 5898 | 1.04 |
| | AMC23 | 85 | 5155 | 1.00 | – | – | 5194 | 2.00 | 5624 | 1.04 | 5638 | 1.04 |
| | MATH500 | 71 | 3426 | 1.00 | – | – | 3434 | 2.00 | 3657 | 1.05 | 3696 | 1.04 |
| 4B | GSM8K | 61 | 1345 | 1.00 | – | – | 1306 | 2.00 | 1564 | 1.06 | 1498 | 1.06 |
| | GPQA | 234 | 6200 | 1.00 | – | – | 6133 | 2.00 | 6106 | 1.12 | 6120 | 1.12 |
| | MMLU-STEM | 97 | 1589 | 1.00 | – | – | 1541 | 2.00 | 1632 | 1.11 | 1577 | 1.11 |
| | HumanEval++ | 133 | 3047 | 1.00 | – | – | 2776 | 2.00 | 3312 | 1.05 | 3030 | 1.09 |
| | MBPP++ | 51 | 2673 | 1.00 | – | – | 2402 | 2.00 | 2633 | 1.05 | 2584 | 1.05 |
| | Average ratio | – | $1.00\times$ | $1.00\times$ | – | – | $0.98\times$ | $2.00\times$ | $1.05\times$ | $1.06\times$ | $1.03\times$ | $1.07\times$ |

oracle discrepancy metric, top-1 mismatch gives the best trajectory under matched continuation budgets, supporting our default policy. Note that these training-scheme dynamics reflect a performance ceiling rather than the actual downstream task performance.

### A.3.2. ITERATION LABEL ANALYSIS

**Iteration label identifiability**. We first examine whether the oracle iteration labels expose stable token-level signals. We compute the token entropy of continue-labeled and stop-labeled tokens across three diverse subsets of the Open-R1 dataset (Math, Science, and Code). As shown in Figure 13, continue-labeled tokens exhibit a universal signature of significantly higher entropy ($> 5\times$) than stop-labeled tokens. This distinct separation confirms that iteration need is an intrinsic, robustly identifiable property of the model's predictive state, rather than a complex, task-specific pattern. Given this clear signal, the neural iteration decider can learn reliable classification strategies that generalize well across different domains.

**Cross-model label consistency**. We next analyze whether iteration labels are stable across reference model scales. Figure 11 shows substantial overlap; for example, the 1.7B reference identifies 81% of the iteration-selected tokens found by the 4B reference. To assess the quality of this overlap, we compare the cross-entropy of overlap and non-overlap iteration-selected

*Table 20.* Input tokens (shared across methods) and output token / iteration statistics for Standard, AlwaysThink, TaH, and TaH+ (Math-trained version).

| | | | Standard | | AlwaysThink | | TaH | | TaH+ | |
|---|---|---|---|---|---|---|---|---|---|---|
| **Param.** | **Dataset** | **In.** | Out. | Iter. | Out. | Iter. | Out. | Iter. | Out. | Iter. |
| | AIME25 | 159 | 7450 | 1.00 | 7316 | 2.00 | 7648 | 1.05 | 7486 | 1.06 |
| | OlympiadBench | 100 | 6599 | 1.00 | 6622 | 2.00 | 6631 | 1.09 | 6513 | 1.06 |
| | AMC23 | 85 | 6377 | 1.00 | 6368 | 2.00 | 6242 | 1.05 | 6145 | 1.05 |
| *0.6B* | MATH500 | 71 | 4823 | 1.00 | 5350 | 2.00 | 4877 | 1.05 | 4793 | 1.06 |
| | GSM8K | 61 | 1955 | 1.00 | 2844 | 2.00 | 1923 | 1.07 | 1791 | 1.07 |
| | Average ratio | – | 1.00× | 1.00× | 1.02× | 2.00× | 1.00× | 1.06× | 0.97× | 1.06× |
| | AIME25 | 159 | 7195 | 1.00 | 7173 | 2.00 | 7496 | 1.06 | 7498 | 1.06 |
| | OlympiadBench | 100 | 6008 | 1.00 | 6484 | 2.00 | 6387 | 1.06 | 6258 | 1.06 |
| | AMC23 | 85 | 5681 | 1.00 | 7543 | 2.00 | 6122 | 1.04 | 5852 | 1.06 |
| *1.7B* | MATH500 | 71 | 4004 | 1.00 | 4414 | 2.00 | 4233 | 1.06 | 4286 | 1.06 |
| | GSM8K | 61 | 1451 | 1.00 | 1644 | 2.00 | 1721 | 1.08 | 1686 | 1.08 |
| | Average ratio | – | 1.00× | 1.00× | 1.13× | 2.00× | 1.09× | 1.06× | 1.07× | 1.06× |

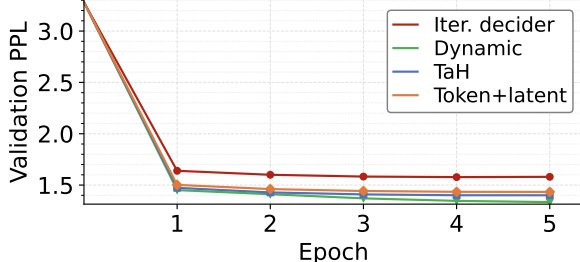

*(a)* Supervision type and iteration policy.

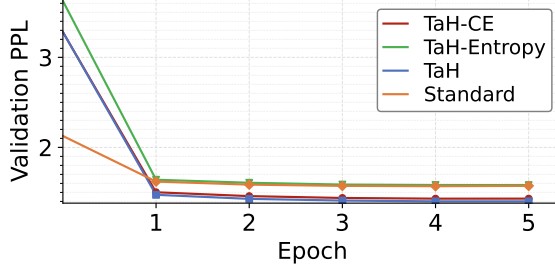

*(b)* Oracle discrepancy metric.

*Figure 10.* Validation-perplexity dynamics for the training-scheme ablations in Table 4. (a) Supervision and iteration-policy choices. (b) Oracle discrepancy metrics under matched continuation budgets.

tokens in Figure 12. Overlap tokens have consistently higher loss ($\approx 2.0\times$), indicating a stable core of genuinely difficult token positions. These results show that iteration labels expose stable uncertainty signals, motivating the generalization analysis of the learned iteration decider below.

### A.3.3. ITERATION DECIDER ANALYSIS

**Cross-domain behavior**. We evaluate the iteration decider, trained on the general Open-R1 corpus, across three validation subsets (Math, Code, and QA) to quantify its robustness and cross-domain generalizability. As summarized in Table 22, the decider maintains high decision accuracy across all domains without any retraining.

Despite being invoked on only 7.8-26.6% of tokens, the decider consistently yields 5.2-9.0% absolute accuracy gains over the standard single-pass baseline on all three domains. Moreover, the decider automatically adjusts its iteration rate according to task difficulty: it iterates more frequently on QA (26.6%) than on Math (7.8%), even under a fixed threshold $c_{\text{threshold}} = 0.9$. This behavior indicates that the decider responds to intrinsic uncertainty signals in the model's predictive distribution rather than memorizing domain-specific patterns.

**Iteration decision error**. We further analyze how specific decision mistakes affect end-to-end response quality. Because the learned decider is imperfect, as shown in Figure 5, we randomly inject errors into the oracle iteration-decider predictions at different rates. Formally, we denote the original oracle prediction as the *label* $l \in \{0, 1\}$ and the altered prediction as the

*Table 21.* Decoding computation (GFLOPs) and memory access (GB) per output token for Standard, AlwaysThink, TaH, and TaH+.

| Param. | Dataset | Standard | | AlwaysThink | | TaH | | TaH+ | |
|---|---|---|---|---|---|---|---|---|---|
| | | Comp. | Mem. | Comp. | Mem. | Comp. | Mem. | Comp. | Mem. |
| *0.6B* | AIME25 | 1.47 | 1.38 | 3.35 | 3.14 | 1.52 | 1.43 | 1.57 | 1.47 |
| | OlympiadBench | 1.41 | 1.32 | 3.21 | 3.02 | 1.51 | 1.42 | 1.50 | 1.41 |
| | AMC23 | 1.40 | 1.31 | 3.17 | 2.97 | 1.43 | 1.34 | 1.46 | 1.37 |
| | MATH500 | 1.31 | 1.22 | 2.98 | 2.80 | 1.35 | 1.26 | 1.39 | 1.31 |
| | GSM8K | 1.14 | 1.06 | 2.54 | 2.38 | 1.19 | 1.12 | 1.22 | 1.14 |
| | Average ratio | 1.00× | 1.00× | 2.27× | 2.27× | 1.04× | 1.04× | 1.06× | 1.06× |
| *1.7B* | AIME25 | 4.31 | 4.03 | 9.45 | 8.83 | 4.51 | 4.21 | 4.64 | 4.34 |
| | OlympiadBench | 4.16 | 3.88 | 9.18 | 8.58 | 4.36 | 4.07 | 4.48 | 4.18 |
| | AMC23 | 4.12 | 3.85 | 9.54 | 8.91 | 4.24 | 3.96 | 4.43 | 4.13 |
| | MATH500 | 3.92 | 3.66 | 8.45 | 7.89 | 4.10 | 3.83 | 4.23 | 3.95 |
| | GSM8K | 3.62 | 3.38 | 7.48 | 6.98 | 3.87 | 3.61 | 3.98 | 3.72 |
| | Average ratio | 1.00× | 1.00× | 2.19× | 2.19× | 1.05× | 1.05× | 1.08× | 1.08× |

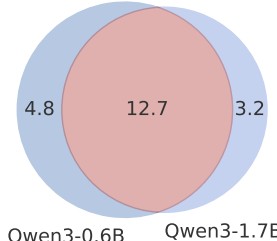

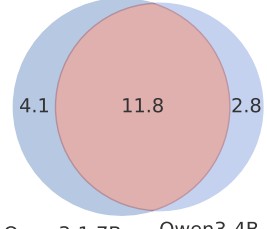

*(a)* Overlap between Qwen3-0.6B and 1.7B      *(b)* Overlap between Qwen3-1.7B and 4B

*Figure 11.* Venn diagrams illustrating the overlap of iteration-selected tokens between different reference models. The high overlap proportions indicate that iteration labels are largely consistent across model scales.

*output* $o \in \{0, 1\}$. We define the *iter. error* as the total proportion of deliberately introduced errors:

$$\text{iter. error} = P(l \neq o) = \underbrace{P(l = 1, o = 0)}_{\text{underthink rate}} + \underbrace{P(l = 0, o = 1)}_{\text{overthink rate}}. \tag{13}$$

We further distinguish the impacts of overthinking and underthinking. Here, overthinking refers to cases where the decider incorrectly signals *continue*, while underthinking corresponds to cases where it incorrectly signals *stop*. Table 23 shows how TaH's MATH100 accuracy varies with different iteration error rates. We quantify these effects by fitting a linear model to the data:

$$\text{accuracy} = -1.41 \times \text{underthink rate} - 2.73 \times \text{overthink rate} + 0.81.$$

This analysis indicates that inaccurate iteration decisions are the main factor behind the performance gap between TaH and its oracle variant, with overthinking being the dominant source of performance gaps.

A.3.4. TOKEN ALTERNATION PATTERN

We analyze tokens that most frequently trigger a second iteration ("think-twice" tokens). For each token type $t$, we compute the continuation rate

$$\Pr\left(c_i^{(1)} > c_{\text{threshold}} \mid t_i = t\right),$$

using the inference threshold $c_{\text{threshold}} = 0.9$ (Section 5.3). We estimate this quantity on the Open-R1 validation set and, for diagnostics, randomly sample 10K token positions ($\approx$0.4% of tokens) to track whether the next-token prediction switches between depth 1 and depth 2. This setting quantifies which token types most often trigger an additional iteration and how often iteration alters the predicted next token.

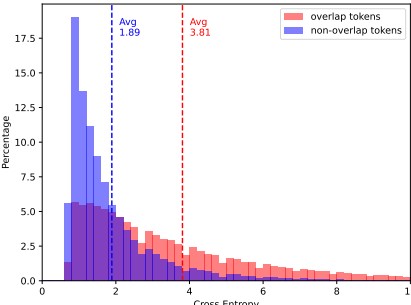

*(a)* Qwen3-1.7B: cross-entropy of overlap vs. non-overlap iteration-selected tokens (w.r.t. Qwen3-0.6B).

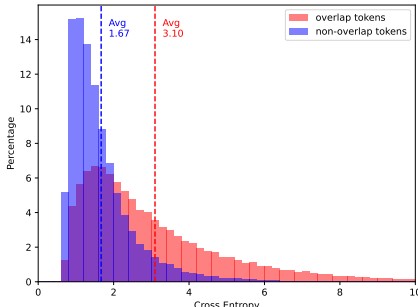

*(b)* Qwen3-0.6B: cross-entropy of overlap vs. non-overlap iteration-selected tokens (w.r.t. Qwen3-1.7B).

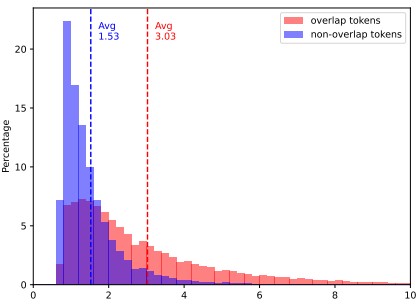

*(c)* Qwen3-4B: cross-entropy of overlap vs. non-overlap iteration-selected tokens (w.r.t. Qwen3-1.7B).

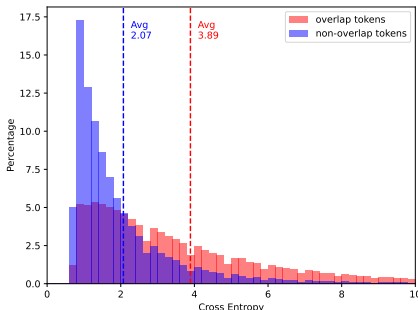

*(d)* Qwen3-1.7B: cross-entropy of overlap vs. non-overlap iteration-selected tokens (w.r.t. Qwen3-4B).

*Figure 12.* Token-level cross-entropy distributions of overlap and non-overlap iteration-selected tokens across different reference model pairs. For each pair of reference models (e.g., Qwen3-1.7B and Qwen3-0.6B), we plot the cross-entropy of tokens selected for iteration by both models (overlap) and by only one model (non-overlap) on both reference models.

### A.3.5. DUO-CAUSAL ATTENTION PATTERN

We perform forward computation on 100 samples, each with a length of 128 tokens, and visualize the learned attention patterns of the TaH model during the second iteration.

**Qualitative analysis**. Figure 14 shows the average attention weights of three representative heads. The left panel illustrates a head that mainly attends to first-iteration keys; the middle panel shows one focusing on second-iteration keys; and the right panel displays a head with balanced attention across both iterations. These examples demonstrate that the duo-causal attention mechanism enables the model to automatically learn diverse cross-depth attention strategies.

**Quantitative analysis**. Figure 15 further quantifies, for each layer, how much attention mass second-iteration queries allocate to first-iteration keys. Following the TaH framework, let $a_{\ell,h}\big((i,d_q) \to (j,d_{kv})\big)$ denote the attention weight at layer $\ell$, head $h$, from the query at position $i$, depth $d_q$ to the key at position $j$, depth $d_{kv}$. As an illustrative example, consider query $(3,2)$ highlighted in Figure 3(c): it attends to three depth-1 keys and two depth-2 keys. We want to quantify how

*Table 22.* Iteration decider behavior and downstream gains on different validation subsets. The decider is trained once on general Open-R1 and evaluated without retraining.

| Metric | Math | Code | QA |
|---|---|---|---|
| Iteration Percentage | 7.8% | 10.7% | 26.6% |
| Iteration Accuracy | 86.7% | 82.3% | 76.6% |
| Benchmark Gain over Standard | **+5.2%** | **+9.0%** | **+5.8%** |

*Table 23.* TaH performance under different iteration-decider error rates. All values are reported in percentages.

| Iter. Error (%) | Underthink (%) | Overthink (%) | MATH100 Acc. (%) |
|---|---|---|---|
| 0.0 | 0.0 | 0.0 | 80.0 |
| 2.8 | 2.8 | 0.0 | 78.0 |
| 10.0 | 1.5 | 8.5 | 55.4 |
| 15.0 | 2.1 | 12.9 | 45.2 |
| 20.0 | 2.5 | 17.5 | 27.1 |
| 22.1 | 0.0 | 22.1 | 21.6 |

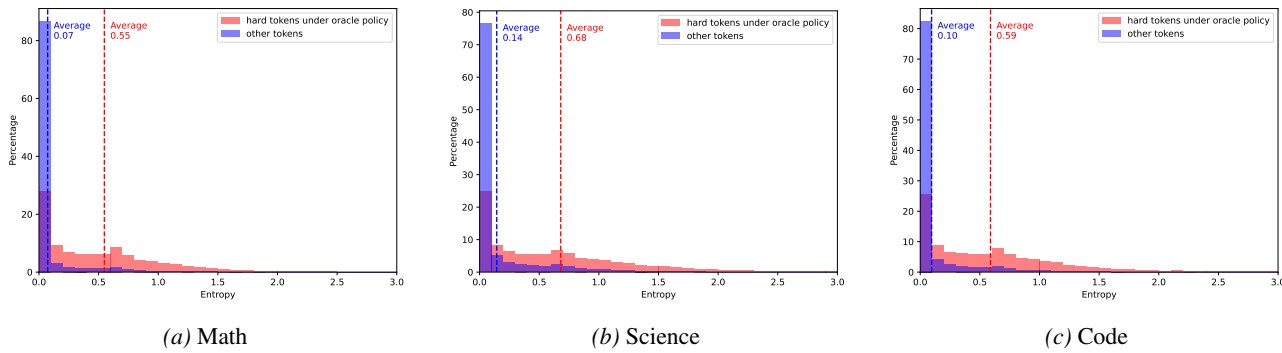

*(a)* Math          *(b)* Science          *(c)* Code

*Figure 13.* Output-logit entropy distribution at the first iteration of TaH, categorized by oracle iteration labels on the Open-R1 validation set (Math, Science, Code). The distinct separation between distributions confirms that TaH's internal logits provide a strong, task-agnostic signal for identifying tokens that require additional iteration.

*Table 24.* Conditional probabilities of continuation confidence and next-token distribution.

| **Token** $T_1$ | $P(c^{(1)} > c_{\text{threshold}} \mid t^{(1)} = T_1)$ | **Token** $T_2$ | $P(t^{(2)} = T_2 \mid t^{(1)} = T_1)$ |
|---|---|---|---|
| But | 34.3% | So | 13.63% |
| | | Wait | 12.17% |
| | | Therefore | 8.95% |
| So | 17.7% | So | 28.17% |
| | | Therefore | 13.67% |
| | | But | 4.89% |

attention mass is distributed between these two groups. We define the *cross-iteration attention mass* for a single query position as

$$m_{i,\ell,h}^{(2 \to 1)} \;=\; \sum_{j \leq i} a_{\ell,h}\big((i,2) \to (j,1)\big),$$

and average over all depth-2 query positions within each sequence $s$:

$$\bar{m}_{s,\ell,h}^{(2 \to 1)} \;=\; \frac{1}{|Q_s^{(2)}|} \sum_{i \in Q_s^{(2)}} m_{i,\ell,h}^{(2 \to 1)},$$

where $Q_s^{(2)}$ is the set of token positions that iterate to depth 2 in sequence $s$. The curve in Figure 15 plots, for each layer, the mean of $\bar{m}_{s,\ell,h}^{(2 \to 1)}$ over all sequence-head pairs $(s, h)$, with shading indicating one standard deviation.

**Findings**. A higher $\bar{m}^{(2 \to 1)}$ indicates that depth-2 queries rely more on first-iteration keys, whereas a lower value indicates greater reliance on same-iteration keys. The results reveal substantial layer-wise heterogeneity: lower layers place relatively less mass on first-iteration keys, while deeper layers show higher and varied cross-depth reuse.

### A.4. Implementation Details

A.4.1. DUO-CAUSAL ATTENTION IMPLEMENTATION

Figure 3 illustrates the implementation of duo-causal attention, with the formal definitions provided below.

**(1) KV cache concatenation.** At depth $d$, we form the visible K/V sequence by concatenating all shallower-to-current depths along the sequence dimension:

$$\text{KV}^{(\leq d)} \;=\; \big[\, \text{KV}^{(1)}; \text{KV}^{(2)}; \cdots; \text{KV}^{(d)} \,\big].$$

This realizes the accessible set in Equation 6, allowing deeper iterations to access all shallower iterations while preserving positional causality. The KV cache is managed by iteration depth during decoding, as shown in Figure 3(b). The fragmented KV-cache management strategy is standard in existing LLM serving systems (Kwon et al., 2023; Zheng et al., 2024).

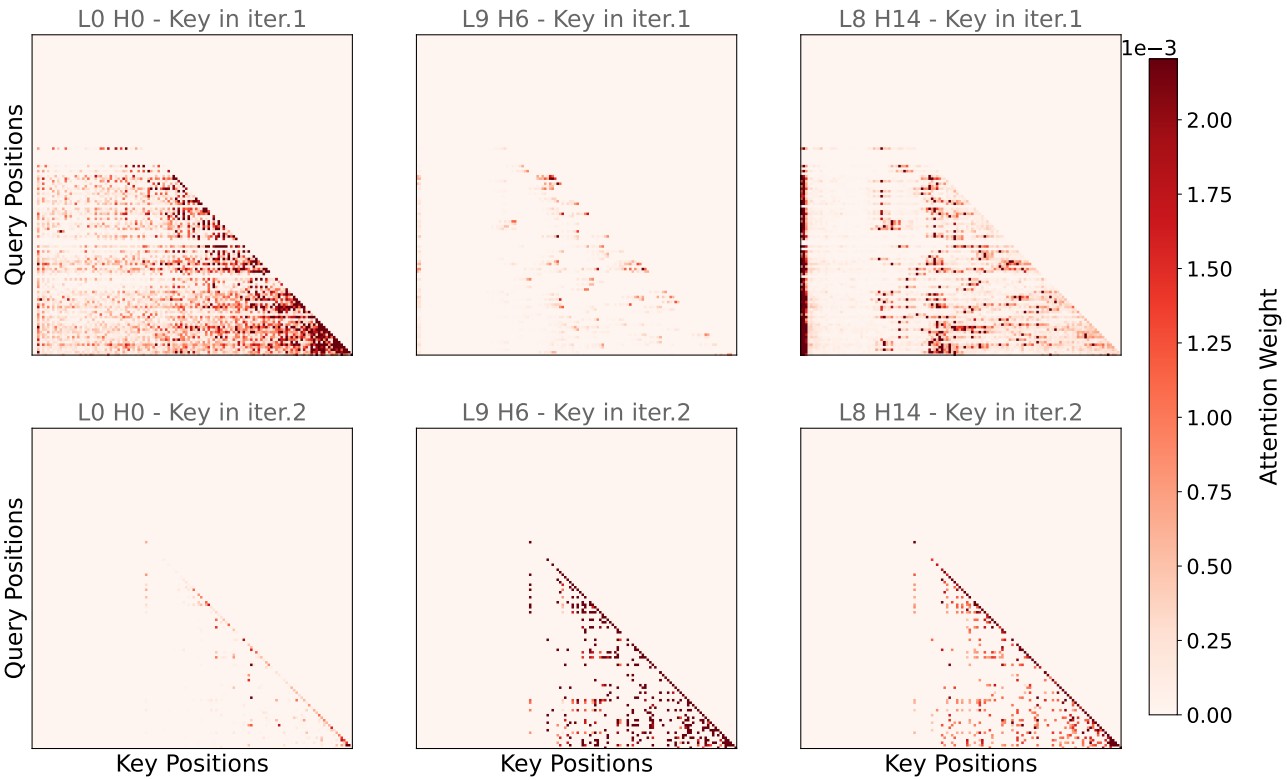

*Figure 14.* TaH duo-causal attention pattern.

**(2) Two-dimensional causal mask.** For a query $(i, d)$, a key $(j, k)$ is attendable iff $j \leq i$ and $k \leq d$. We implement this as an additive attention mask with $0$ for allowed entries and $-\infty$ otherwise, enforcing positional and iteration causality jointly. Figure 3(c) visualizes the landscape of the duo-causal attention mask. When $d = 1$ for all tokens, the rule reduces to standard causal attention.

**(3) Compatibility with efficient attention.** The mask is provided in the standard additive form and the concatenated K/V tensors remain contiguous along the sequence dimension, matching the usual scaled dot-product attention interface. As a result, duo-causal attention is directly compatible with optimized kernels such as FlashAttention, without kernel modifications.

### A.5. Additional Related Work

Instead of using the shared model parameters multiple times through latent iteration, previous work also proposes layer skipping methods for dynamic compute allocation.

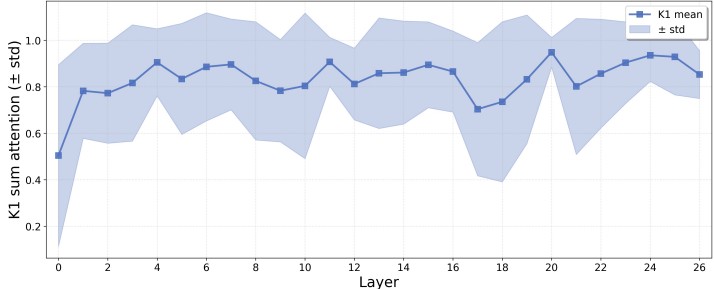

*Figure 15.* TaH layer-wise cross-iteration attention mass $\bar{m}^{(2 \to 1)}$ (mean $\pm$ one standard deviation over sequences and heads).

**Layer skipping**. Layer skipping aims to accelerate LLM inference by dynamically bypassing certain layers for specific tokens. Some methods use a learnable module to make real-time skipping decisions. MoD (Raposo et al., 2024) uses a top-k router to select a subset of tokens for processing, while FlexiDepth (Luo et al., 2025) uses a plug-in router to determine whether a layer should be bypassed. Others use a fixed strategy to skip layers. SkipDecode (Del Corro et al., 2023) enforces a monotonically decreasing number of active layers during generation. However, these methods still require loading the entire model's parameters, resulting in a large memory access overhead. Instead of skipping some layers, TaH adds computational depth by allowing core tokens to undergo multiple refinement iterations. This approach provides greater computational depth without increasing the model's parameter count.

## A.6. Limitations and Future Work

**Comparison with official Qwen3 models**. Official Qwen3 models are trained on different data distributions and scales, and use different training procedures, including on-policy distillation (Yang et al., 2025). By contrast, our models use SFT only on limited, publicly accessible data. Consequently, performance may differ between the two.

**Future work**. We consider advanced training techniques largely orthogonal to TaH. Promising directions include (1) integrating TaH with online distillation to provide stronger supervision; (2) applying reinforcement learning to optimize the compute–accuracy trade-off and to learn continuation policies beyond imitation. Due to resource constraints, this work focuses on SFT with public data; we leave these combinations for future work.

## A.7. Use of Large Language Models (LLMs)

Large language models (e.g., GPT-5, Gemini) were used exclusively for proofreading and minor copyediting. They were not employed to generate scientific content, contribute to research ideation, or design methods. All research ideas, methods, and conclusions are solely the responsibility of the authors.

## A.8. Examples

We use an example question from AIME25 and responses from Standard, TaH and AlwaysThink to provide an intuitive understanding of our method. Only TaH can successfully answer this question. Blue-highlighted tokens indicate positions that undergo two iterations.

> **Text 1. Question**
>
> In $\triangle ABC$, points $A, D, E,$ and $B$ lie in that order on side $\overline{AB}$ with $AD = 4, DE = 16,$ and $EB = 8$. Points $A, F, G,$ and $C$ lie in that order on side $\overline{AC}$ with $AF = 13, FG = 52,$ and $GC = 26$. Let $M$ be the reflection of $D$ through $F$, and let $N$ be the reflection of $G$ through $E$. Quadrilateral $DEGF$ has area 288. Find the area of the heptagon $AFNBCEM$.

## Text 2. Standard Output

Okay, let me try to figure out this problem step by step. It's about a heptagon formed by reflecting points and finding its area. ... We have triangle $ABC$. On side $AB$, points $A, D, E$, and $B$ are in that order. The lengths are given as $AD = 4$, $DE = 16$, and $EB = 8$. So $AB = 28$ ... On side $AC$, points $A, F, G$, and $C$ are in that order with $AF = 13$, $FG = 52$, and $GC = 26$. So $AC = 91$ ... Let me place point $A$ at $(0,0)$ and $B$ at $(28,0)$ ... Let $C$ be at $(91, h)$ ... Then $D = (4, 0)$, $E = (20, 0)$, $F = \left(13, \frac{13h}{91}\right)$, $G = \left(65, \frac{65h}{91}\right)$ ...
Area of quadrilateral $DEGF$ can be calculated ... Applying the shoelace formula ... Area $= \frac{1}{2} \cdot \frac{1248h}{91} = \frac{624h}{91} = 6h$
So with 288: $6h = 288 \Rightarrow h = \frac{288}{6} = 48$
Reflections: $M$ is the reflection of $D$ through $F$ and $N$ is the reflection of $G$ through $E$ ... $M = (22, 0)$, $N = (-39, 0)$, $B = (28, 0)$, $C = (91, 48)$, $E = (20, 0)$ ...
Using the shoelace formula for $AFNBCEM$ in order ... Sum $= 0 + 0 + 0 + 1344 + (-960) + 0 + 0 = 384$
Take the absolute value and multiply by 1/2: Area $= \frac{1}{2} \cdot |384| = 192$
...
**Final Answer** $\boxed{192}$ ✗

## Text 3. TaH Output

Okay, let's try to tackle this geometry problem. It seems a bit complex with all these points and reflections, ... First, let me visualize triangle ABC. Points D, E, and B are on side AB with AD = 4, DE = 16, and EB = 8. ... Let me set up a coordinate system. Let's place point A at the origin $(0,0)$ ... Then, since $AB$ is a line segment, let me assume $AB$ is along the $x$-axis. So point $B$ would be at $(28, 0)$ ... Now, points $F, G, C$ are on $AC$ ... Let me define coordinates for $C$ as $(c_x, c_y)$ ...
...
Compute the sum of $x_i y_{i+1} - x_{i+1} y_i$ for each $i$ ...
Sum of all terms: $0 + \frac{1300c_y}{91} + 0 -> \frac{52c_y}{91} = \frac{(1300-52)c_y}{91} = \frac{1248c_y}{91}$
Then, the area is $\frac{1}{2} \cdot \left| \frac{1248c_y}{91} \right| = \frac{624c_y}{91}$
...
Compute the area of this heptagon ...
Using the shoelace formula ...
Term2: –240   Term3: 840
Term4: 1176   Term5: –840
Term6: 240 ...
Sum: $0 - 240 + 840 + 1176 - 840 + 240 + 0 = 1176$
Take the absolute value and multiply by 1/2: Area $= \frac{1}{2} \cdot |1176| = 588$
...
**Final Answer** $\boxed{588}$ ✓

## Text 4. AlwaysThink Output

Okay, so I need to find the area of the heptagon AFNBCEM. Let me first try to visualize the problem. There's triangle ABC. On side AB, there are points A, D, E, and B. The order is A, D, E, B. The distances given are AD=4, DE=16, and EB=8. So, starting at A, moving 4 units to D, then 16 units to E, then 8 units to B. So, the total length of AB is AD + DE + EB = 4 + 16 + 8 = 28 units. Got that.
...
Points in order: D(4,0), E(20,0), F(13,h), G(65,h), back to D(4,0).
Compute the terms:
...
Sum these terms: $0 + 20h - 52h - 4h = -36h$
Take the absolute value and multiply by 1/2: $Area = (1/2) * | -36h| = (1/2) * 36|h| = 18|h|$
But the area is given as 288, so $18|h| = 288 \Rightarrow |h| = 16$.
...
Now, apply the shoelace formula ... Therefore, total sum is $8640/17$
Take the absolute value and multiply by 1/2: $Area = (1/2) * |8640/17| = 8640/34 = 4320/17 \approx 254.1176$
...
Therefore, the answer is $4320/17$.

**Final Answer** $\boxed{\dfrac{4320}{117}}$ ✗

