# OpenReview forum: "Think-at-Hard: Selective Latent Iterations to Improve Reasoning Language Models"
_ICML.cc/2026/Conference — ICML 2026 regular_

### Official Review · Reviewer_RduM · 2026-03-12

**Soundness:** 3
**Presentation:** 3
**Significance:** 3
**Originality:** 3
**Overall Recommendation:** 5
**Confidence:** 3

**Summary:**

This paper proposes TaH (Think-at-Hard), a method for improving reasoning efficiency in looped transformer via selective latent iterations. The key idea is that the model performs additional hidden-state iterations only for tokens that are predicted to be difficult, while easy tokens are generated directly without extra computation. To enable this mechanism, the authors introduce a duo-causal attention design that allows tokens at deeper reasoning depths to access shallower states of previous tokens. The model further distinguishes between prediction (depth = 1) and correction (depth > 1) using LoRA adapters, and employs a lightweight MLP-based decider to determine whether further iterations are necessary. Experiments on 9 reasoning benchmarks show that TaH achieves 3–6.2% accuracy improvements with only 4–5% additional computation, outperforming several baselines.

**Compliance With Llm Reviewing Policy:**

Affirmed.

**Key Questions For Authors:**

Could the authors provide additional analysis on weakness 1 and 2？

**Limitations:**

Yes，

**Strengths And Weaknesses:**

## Strengths：
1. The work try to address the “overthinking” issue in looped transformers, where iterative computation is applied to all tokens regardless of difficulty. The "oracle" experiment in Section 4 provides compelling evidence for the potential of selective iteration: skipping iterations for 81–88% of tokens not only reduces computation but also improves performance by 2.1%. This observation provides strong motivation for the proposed approach.
2. The overall design is well-structured. The duo-causal attention mechanism enables cross-depth context sharing, while LoRA adapters separate the roles of prediction and correction during different reasoning depths. The MLP decider provides a lightweight mechanism for dynamically selecting computation depth. The design is intuitive and aligns well with the underlying motivation.
3. The authors evaluate TaH on 9 benchmarks spanning math, code, and scientific QA, across three model scales (0.6B, 1.7B, and 4B). The paper reports multiple efficiency metrics including accuracy, compute overhead, memory usage, and latency, demonstrating that the method consistently improves performance while maintaining low computational overhead.

## Weaknesses:
1. In Section 5.3, the decider is trained using an oracle policy derived from an SFT model. As a result, the effectiveness of the decider may heavily depend on the quality and behavior of the SFT model. The paper does not provide sufficient analysis of how sensitive the learned policy is to the training data or to potential biases in the SFT model’s outputs. More analysis would strengthen the empirical claims.
2. Table 13 shows that the model rarely chooses iteration depth 3. It is unclear whether this phenomenon arises from the training data distribution, the oracle policy used to supervise the decider, or limitations of the proposed mechanism itself. This raises questions about the scalability of the approach when deeper reasoning iterations are required. Additional analysis or ablations could help clarify this issue.

---

> ### Author Rebuttal · Authors · 2026-03-31
>
> We sincerely thank Reviewer RduM for the strong recognition. We appreciate your valuable appreciation of our **motivation, architecture design, and comprehensive evaluation**. We address the concerns below.
>
> ---
> ### W1: Decider sensitivity to SFT model
>
> > In Section 5.3, the decider is trained using an oracle policy derived from an SFT model. ... The paper does not provide sufficient analysis of how sensitive the learned policy is to the training data or to potential biases in the SFT model’s outputs.
>
> We appreciate the reviewer's concern and address it with the following analysis.
>
> **Role of the SFT model.** We wish to note that the reference SFT model acts as a failure detector, not a teacher. It simply marks model-specific failures and does not need to align with human perception of difficulty.
> Compared with prior works that randomly sample iteration depths without condition on per-token difficulty [1], TaH's oracle targets tokens that the model fails on, enabling more focused refinement.
>
> **Empirical Verification** We study whether different SFT model brings biases that may corrupt the ocacle policy.
> Specifically, we show that even a smaller, "inaccurate" reference SFT model (1.7B) successfully matches 81% of a larger (4B) model's "hard" labels (Appendix A.3.1). The identified labels correspond to the 4B model's highest uncertainty ($\approx 2.0\times$ higher loss), confirming the robustness of oracle policy to reference model.
>
> [1] Geiping, Jonas, et al. "Scaling up Test-Time Compute with Latent Reasoning: A Recurrent Depth Approach." NeurIPS'25
>
> ---
> ### W2: Infrequent selection of depth 3
>
> > Table 13 shows that the model rarely chooses iteration depth 3. It is unclear whether this phenomenon arises from the training data distribution, the oracle policy used to supervise the decider, or limitations of the proposed mechanism itself. This raises questions about the scalability of the approach when deeper reasoning iterations are required.
>
> We appreciate the careful observation. The low frequency of selecting depth 3 arises primarily from the **oracle policy during training** and the **continuation threshold during inference**. This is a deliberate design choice rather than an inherent limitation of TaH.
>
> **What decides iteration depths distribution.**
> In TaH-$D$ ($D$>2), the iteration policy maps each token to a 1D difficulty axis, from easy to hard.
> During training, the oracle policy assigns iteration depths by cutting this axis into intervals with empirical quantile function (Equation 9). Therefore, the proportion of each depth label is fully configurable.
> During inference, the $c_\text{threshold}$ controls how confident the iteration decider must be before a token iterates further (higher threshold -> fewer iterations).
>
> **Why TaH chooses to decrease iteration frequency.**
> The decreasing frequency of deeper iterations is by design, reflecting the long-tailed distribution of token difficulty. In response to the reviewer's concern, we compare our decreasing setup with a uniform setup:
> * Decreasing: training with iteration depths 0.84 : 0.08 : 0.08, inference with $c_\text{threshold}=0.9$. 0.84 and 0.9 inherited from TaH-2 setting.
> * Uniform:  training with iteration depths 0.34 : 0.33 : 0.33, inference with $c_\text{threshold}=0.5$.
>
> | Bench | Standard | TaH-2-Decreasing | TaH-3-Uniform | TaH-3-Decreasing | TaH-4-Decreasing |
> |---|---:|---:|---:|---:|---:|
> | MATH500 | 68.4 | 74.4 [5.6] | 70.2 [29.4, 31.8] | 74.8 [6.5, 1.2] | 74.8 [4.9, 0.6, 0.6] |
> | GSM8K | 82.1 | 84.5 [7.5] | 82.5 [29.5, 35.8] | 84.0 [8.6, 1.8] | 84.6 [7.1, 0.9, 0.8] |
> | AMC23 | 42.2 | 48.4 [4.2] | 41.6 [30.7, 32.9] | 49.1 [6.3, 1.0] | 49.7 [4.3, 0.5, 0.4] |
> | Olympiad | 33.0 | 38.8 [5.7] | 34.4 [30.6, 34.1] | 41.6 [5.3, 1.0] | 40.7 [4.0, 0.5, 0.4] |
> | AIME25 | 13.3 | 17.9 [6.0] | 14.2 [30.4, 35.3] | 19.6 [5.4, 1.0] | 20.4 [4.3, 0.5, 0.3] |
> | Avg | **47.8** | **52.8** [5.8] | **48.6** [30.1, 34.0] | **53.8** [6.4, 1.2] | **54.0** [4.9, 0.6, 0.5] |
>
> > Entries formatted as `Acc. [%iter2, %iter3, %iter4]`
>
> The decreasing setup consistently outperforms the uniform setup, confirming that sparser allocation of deeper iterations is more effective than forcing frequent use of depth 3.
>
> We also identified and fixed a minor bug in the original TaH-3-Decreasing decider in Table 13; the updated results above show improved accuracy. We will update the paper accordingly.
>
> **Scalability to deeper iterations.**
> As shown in the table above, TaH-3-Decreasing and TaH-4-Decreasing achieve average gains of +1.0% and +1.2% over TaH-2, respectively, demonstrating that the framework consistently benefits from additional depth.

---

> > ### Author Rebuttal · Reviewer_RduM · 2026-04-07
> >
> > Thank you for your response. I agree that the positive score is reasonable.

---

### Official Review · Reviewer_YrK4 · 2026-03-13

**Soundness:** 3
**Presentation:** 3
**Significance:** 3
**Originality:** 4
**Overall Recommendation:** 4
**Confidence:** 3

**Summary:**

This work identifies a phenomenon referred to as latent overthinking in looped transformer architectures, where tokens that are correctly reasoned about at earlier depths may become incorrect at later depths. The authors propose a model architecture and training strategy to mitigate this issue. Specifically, the method introduces depth-wise attention mechanisms and depth-specific LoRA weights, enabling the model to exhibit different behaviors across depths while still effectively leveraging information accumulated through multiple iterations. As a result, compared to prior approaches that simply maximize the number of iterations, the proposed method achieves consistent performance improvements across several benchmarks.

**Compliance With Llm Reviewing Policy:**

Affirmed.

**Final Justification:**

The rebuttal effectively addresses my primary concern regarding scalability, providing both empirical evidence and a plausible explanation for the reduced gains at larger model sizes. In addition, the follow-up experiments on cross-scale and noisy labeling further strengthen the practical robustness of the proposed method. Overall, the clarifications reinforce my initial positive assessment, and I maintain my original recommendation.

**Key Questions For Authors:**

- How does performance change when models of different sizes are used for labeling? For example, if a 0.6B model is used for labeling while a 4B model is trained, does this lead to performance degradation? If not, such a setup could significantly reduce the computational cost of the labeling process.

- In the duo-causal attention mechanism, do keys from the previous depth and the current depth receive comparable attention weights, or does the model learn different weighting patterns across depths? In other words, is explicit depth-wise weight tuning necessary?

- Does the method still provide performance improvements when the maximum depth used during training and inference differs? For example, if the model is trained with a maximum depth of 3 but uses a maximum depth of 1 or 2 during inference, does the improvement trend still hold?

**Limitations:**

yes

**Strengths And Weaknesses:**

### Strengths
- The paper clearly formulates the problem of latent overthinking, which appears to be an important issue in looped transformer architectures. The authors demonstrate this phenomenon through intuitive oracle experiments, showing that only a small subset of tokens actually requires additional refinement.

- The paper proposes an intuitive architectural and training solution to address latent overthinking. Under the key hypothesis that only an appropriate depth should be used for each token, the method introduces residual connections and attention mechanisms that aggregate information from tokens that remain under-refined at earlier depths. In addition, the paper proposes a model-based labeling strategy to generate depth-wise ground-truth labels.

- The paper provides strong ablation studies. By removing individual components or replacing them with alternatives, the authors demonstrate that each component of the proposed method plays an important role in mitigating latent overthinking.

- The method is computationally efficient compared to competing approaches. Since only a subset of tokens requires deeper forwarding, the method achieves faster inference, which is practically important.

### Weaknesses
The performance gain appears to become more marginal as model size increases. For example, in the 4B model, the improvement over the standard model is limited to approximately 2.2 points. When compared with the 0.6B and 1.7B models, the results suggest that the performance gain decreases as model size grows. It would therefore be important to verify whether the proposed method remains effective at larger scales (e.g., 10B+ models).

---

> ### Author Rebuttal · Authors · 2026-03-31
>
> We sincerely thank Reviewer YrK4 for the positive feedback. We appreciate your valuable recognition of our **problem formulation, method design, and experiment results**. We address the concerns below.
>
> ---
> ### W1: Less performance gain for larger models
>
> > The performance gain appears to become more marginal as model size increases... It would therefore be important to verify whether the proposed method remains effective at larger scales (e.g., 10B+ models).
>
> We appreciate the reviewer's observation. We believe the reduced absolute gains at 4B reflect a ceiling effect rather than a scalability limitation, as we detail below.
>
> **TaH gains are larger on harder tasks.** Since the 4B baseline already exceeds 85% on several benchmarks, headroom for absolute gains is naturally limited. TaH-4B still brings meaningful gains on harder benchmarks, e.g., **+2.9** on AIME25 (24.2 → 27.1) and **+5.5** on AMC23 (64.2 → 69.7). More broadly, across the 9 benchmarks, there is a negative Spearman correlation (r=−0.83, p=0.009 for TaH+) between the Standard baseline score and TaH's absolute improvement, validating more gains on harder tasks.
>
> **4B results may underestimate TaH.** As noted in Table 2, the 4B models were trained with a 4K context cut-off (vs. 8K for smaller models) due to resource limits. Despite this disadvantage, TaH+ still achieves an average improvement of 2.2 points.
>
> We view these results as evidence that TaH effectively improves performance at a larger scale, especially on even more challenging tasks. Since TaH's primary focus is on parameter-constrained edge scenarios, we leave scaling to 10B+ as future work beyond the limited rebuttal period.
>
> ---
> ### Q1: Can we label with small LLM to train large LLM
>
> > if a 0.6B model is used for labeling while a 4B model is trained, does this lead to performance degradation? If not, such a setup could significantly reduce the computational cost of the labeling process.
>
> **Cross-scale labels are largely consistent.** Appendix A.3.1 shows that labels are robust across scales: Qwen3-1.7B identifies 81% of the hard tokens labeled by Qwen3-4B (Figure 8), suggesting potential for cross-scale labeling.
>
> **However, labeling cost is already minimal.** Labeling requires only a one-time offline forward pass over the training corpus, costing less than 10% of the SFT budget. Given this low overhead, we recommend prioritizing label quality with same-scale references. That said, developing a reference-model-free pipeline is an interesting direction for future work.
>
> ---
> ### Q2: Model learns attention weighting across depths?
>
> > In the duo-causal attention mechanism, do keys from the previous depth and the current depth receive comparable attention weights, ...? In other words, is explicit depth-wise weight tuning necessary?
>
> The model automatically learns different weighting patterns across iteration depths, varying by attention head.
>
> In Appendix A.3.6, we provide both qualitative and quantitative analyses of attention weights distribution across iterations. As visualized in Figure 12, three dominant patterns emerge across different heads: (1) heavily attending to iter-1 keys; (2) heavily attending to iter-2 keys; (3) balanced attention across iterations.
> Figure 13 further reveals substantial layer-wise heterogeneity for attention weights distribution.
> These learned specialization patterns confirm that the model naturally discovers depth-wise weighting strategies through our training, without requiring explicit tuning.
>
> ---
> ### Q3: Performance still improves with mismatched maximum depths?
>
> > Does the method still provide performance improvements when the maximum depth used during training and inference differs? ...
>
> Yes, the performance improvements hold even when the inference depth is reduced below the training maximum. We evaluate the TaH-1.7B model, trained with maximum depth 2, under restricted maximum depth 1 during inference:
>
> |Method|Avg.|AIME25|Olympiad|AMC23|MATH500|GSM8K|GPQA|MMLU|HE++|MBPP++|
> |---|:--|:--:|:--:|:--:|:--:|:--:|:--:|:--:|:--:|:--:|
> |Standard|47.5|10.8|33.8|39.7|67.8|80.2|30.3|74.1|39.0|51.9|
> |TaH|**51.3** (+3.8)|13.8|37.2|40.9|71.4|84.8|33.3|74.8|50.0|55.3|
> |TaH (iter1)|**49.6** (+2.1)|13.3|36.3|39.1|69.4|82.2|32.3|75.5|45.4|53.0|
> |TaH+|**53.7** (+6.2)|15.4|37.6|48.4|72.6|84.5|39.4|76.6|51.5|57.5|
> |TaH+ (iter1)|**50.6** (+3.1)|13.3|37.2|45.3|70.6|84.4|29.0|75.8|43.3|56.3|
>
> Although reducing depth degrades performance compared to full TaH, the depth-restricted models still outperform the Standard baseline. Similar to multi-token prediction [1], we hypothesize that TaH's iterative training encourages richer internal representations in the LLM backbone, yielding benefits even without looping at inference. We will expand this discussion in the final paper.
>
> [1] Gloeckle, et al. "Better & Faster Large Language Models via Multi-token Prediction." ICML'24

---

> > ### Author Rebuttal · Reviewer_YrK4 · 2026-04-02
> >
> > We thank the authors for the detailed and thoughtful rebuttal.
> >
> > Most of my concerns have been adequately addressed, and I appreciate the additional analyses and clarifications provided, particularly regarding scaling behavior and depth mismatch.
> >
> > That said, while the responses strengthen the paper, they do not substantially change my overall assessment of its impact and scope. I therefore maintain my original score.

---

> > > ### Author Response · Authors · 2026-04-05
> > >
> > > We thank the reviewer for the thoughtful follow-up and for the positive feedback on our rebuttal. To **further address Q1** on training a larger model with oracle labels from a smaller reference model, we ran two additional experiments that we were unable to complete before the discussion period.
> > >
> > > Specifically, we test end-to-end performance of: (1) using a smaller Qwen3-0.6B as the reference model to generate oracle labels for training TaH-1.7B; and (2) injecting 10% random noise into the Qwen3-1.7B oracle labels (randomly flipping 10% of hard/easy assignments). All experiments use the same TaH-1.7B training recipe and evaluation protocol as in our ablation section.
> > >
> > > | Model | Standard-1.7B | TaH-1.7B | TaH-1.7B | TaH-1.7B |
> > > |:---|:--|:--|:--|:--|
> > > | **Training label** | **None** | **Qwen3-0.6B** | **Qwen3-1.7B (10% noise)** | **Qwen3-1.7B** |
> > > | MATH500 | 68.4 | 70.2 | 73.6 | 74.4 |
> > > | GSM8K | 82.1 | 82.3 | 83.5 | 84.5 |
> > > | AMC23 | 42.2 | 43.4 | 46.8 | 48.4 |
> > > | Olympiad | 33.0 | 33.5 | 36.0 | 38.8 |
> > > | AIME25 | 13.3 | 14.2 | 14.6 | 17.9 |
> > > | **Avg** | **47.8** | **48.7** | **50.9** | **52.8** |
> > >
> > > Even when trained with oracle labels from a smaller 0.6B reference model, TaH-1.7B still improves over Standard-1.7B, as does the noisy label setup. This suggests that TaH is **robust to imperfect oracle labels**, while our standard labeling method still provides the most gains. More broadly, these results suggest that moderate label mismatch will not bottleneck TaH's scaling to larger model sizes. Since label generation is a one-time offline preprocessing step with less than 10% training overhead, we still recommend the original same-scale labeling strategy in practice.
> > >
> > > We hope these additional results further address the reviewer’s concern and strengthen confidence in TaH's practical applicability.

---

### Official Review · Reviewer_yJ5W · 2026-03-13

**Soundness:** 3
**Presentation:** 3
**Significance:** 3
**Originality:** 3
**Overall Recommendation:** 5
**Confidence:** 4

**Summary:**

This paper studies the inefficiency of latent iteration in looped transformers and identifies a latent overthinking phenomenon, where most tokens are already correctly predicted after the first pass but may be unnecessarily revised in later iterations. To address this, the authors propose Think-at-Hard (TaH), a selective iteration framework that triggers additional reasoning only for tokens likely to be incorrect, using a lightweight neural decider together with depth-aware LoRA refinement and a duo-causal attention mechanism that enables cross-iteration information flow. Experiments across multiple reasoning benchmarks show consistent improvements over both single-pass and always-iterate baselines while skipping iterations for the majority of tokens and maintaining minimal computational overhead.

**Compliance With Llm Reviewing Policy:**

Affirmed.

**Final Justification:**

The rebuttal has solved my questions and I support to accept.

**Key Questions For Authors:**

1. I am curious whether the proposed method can demonstrate consistent improvements on at least one non-math benchmark (e.g., GPQA-diamond, CommonsenseQA, or code-related tasks) under the same training setup. Such results would help clarify whether the approach generalizes beyond mathematical reasoning.

2. I am interested in whether the effectiveness of the decider and selective iteration mechanism remains stable across different task domains. Additional ablation studies on non-math benchmarks would help better understand the robustness of the proposed design.

3. I am curious whether a Standard-pruned baseline with the same effective depth as TaH (after pruning) could be provided. This comparison would help more clearly isolate the effect of the selective iteration mechanism itself.

**Limitations:**

see Key Questions and Weaknesses of Originality

**Strengths And Weaknesses:**

Soundness:
The work appears technically sound. The paper identifies the phenomenon of latent overthinking in latent iteration models and proposes a selective iteration framework to address it. The duo-causal attention mechanism enabling depth-wise information flow and the two-stage decoupled training strategy are both reasonable design choices.

Presentation: The paper is well written and generally easy to follow.

Significance:
The paper highlights an interesting and practically relevant issue in latent reasoning models and proposes a selective reasoning strategy to mitigate it. The empirical results demonstrate consistent improvements, suggesting that the approach may offer useful insights for designing more efficient reasoning mechanisms in LLMs.

Originality:
I have some reservations regarding originality. While the idea of selectively applying additional reasoning iterations is intuitive and the design is well executed, the method relies heavily on a supervised oracle policy derived from ground-truth tokens and a frozen SFT reference model, which may limit applicability in settings without high-quality supervision or when the reference model itself is biased. In addition, the experimental evaluation focuses primarily on mathematical reasoning benchmarks, with only limited cross-domain validation, making the claim of domain-agnostic behavior less convincing. Further experiments on a broader range of reasoning tasks would strengthen the paper.

---

> ### Author Rebuttal · Authors · 2026-03-31
>
> We sincerely thank Reviewer yJ5W for the positive feedback. We appreciate your valuable recognition of our **soundness, presentation, and significance**. We address the concerns below.
>
> ---
> ### W1: Reliance on oracle policy
>
> > the method relies heavily on a supervised oracle policy ... limit applicability in settings without high-quality supervision or when the reference model itself is biased.
>
> We address it from three aspects:
>
> **Standard supervision data.** TaH does not impose additional data constraints beyond standard SFT data. All methods use the exact same SFT dataset (Open-R1). TaH's performance gains are from its effective data utilization, without relying on higher-quality data.
>
> **Standard reference model.** The reference model is simply a standard SFT-trained variant of the same base LLM, readily available in standard training pipelines. It only serves as a model-specific failure detector and does not require unbiased alignment with human judgments. Compared with prior works that randomly sample iteration depths without conditioning on per-token difficulty [1], TaH's oracle is more guided, enabling more focused refinement.
>
> **Empirical robustness.** Even a smaller reference model (1.7B) matches 81% of a larger (4B) model's hard-token labels, and the overlapping tokens exhibit ~2.0x higher loss. See Appendix A.3.1 for more details.
>
> [1] Geiping, et al. "Scaling up Test-Time Compute with Latent Reasoning." NeurIPS'25
>
> ---
> ### W2 & Q1: Add evaluation beyond math
>
> > evaluation focuses primarily on mathematical reasoning benchmarks ...
> > I am curious whether the proposed method can demonstrate consistent improvements on at least one non-math benchmark ...
>
> We wish to clarify that our evaluation is **not limited to math**. In the main paper (Table 2), TaH is broadly evaluated across **9 benchmarks** spanning **math (5), QA (2), and coding (2)**, across **6 methods**, all under a **single training** on Open-R1. We average the per-domain results from Table 2 below:
>
> |Size|Method|Math|Code|QA|
> |:--|:--|:--|:--|:--|
> |**0.6B**|Standard|27.6|22.8|42.7|
> ||SoftThink|26.9|21.9|38.9|
> ||Ouro|26.0|21.2|44.7|
> ||AlwaysThink|25.2|11.5|41.1|
> ||TaH|31.0 (+3.4)|27.8 (+5.0)|42.7 (+0.0)|
> ||TaH+|33.9 (+6.3)|28.6 (+5.8)|45.2 (+2.5)|
> |**1.7B**|Standard|46.5|45.5|52.2|
> ||SoftThink|44.2|46.2|53.4|
> ||Ouro|46.1|43.2|52.6|
> ||AlwaysThink|43.4|21.0|50.1|
> ||TaH|49.6 (+3.1)|52.7 (+7.2)|54.1 (+1.9)|
> ||TaH+|51.7 (+5.2)|54.5 (+9.0)|58.0 (+5.8)|
>
> TaH also shows clear gains on non-math reasoning tasks, confirming its strong generalizability. We will clearly highlight the domain coverage of benchmarks in the revision.
>
> ---
> ### Q2: Iteration decider beyond math
>
> > ... the decider and selective iteration mechanism remains stable across different task domains? Additional ablation studies on non-math benchmarks
>
> Iteration decider robustness across domains is detailed in Appendix A.3.2.
>
> **Decider generalizability.**
> The decider is trained once on general Open-R1, then evaluated on its math, code, and QA validation splits:
>
> |Metric|Math|Code|QA|
> |:--|:-:|:-:|:-:|
> |Iter. Rate %|7.8|10.7|26.6|
> |Iter. Acc. %|86.7|82.3|76.6|
> |TaH-1.7B+ End-to-End Gain %|+5.2 |+9.0 |+5.8 |
>
> The decider maintains high accuracy across all domains. It also automatically adjusts iteration rates by task difficulty under a fixed threshold of 0.9.
>
> **OOD generalization.**
> Training *only* on math, TaH still outperforms baselines on general science (MMLU-STEM).
>
> |Benchmark|Standard|SoftThink|AlwaysThink|TaH+|
> |:--|:-:|:-:|:-:|:-:|
> |MMLU-STEM (0.6B)|51.6|51.4|42.6|56.3|
> |MMLU-STEM (1.7B)|70.8|70.6|63.8|73.7|
>
> It confirms that the decider captures a general difficulty signal rather than a math-specific heuristic.
>
> **Why is decider robust?**
> As analyzed in Appendix A.3.3 (Figure 10), hard tokens exhibit a universal signature of >5× higher entropy than easy tokens across tasks. This distinct separation indicates that "hardness" is an intrinsic property of the model's predictive state rather than a task-specific pattern, consistent with findings in [2]. This clear signal enables the strong robustness of decider across domains.
>
> [2] Fu, et al. “R2R: Efficiently Navigating Divergent Reasoning Paths with Small-Large Model Token Routing.” NeurIPS'25
>
> ---
> ### Q3: Comparison with Standard-pruned baseline
>
> > a Standard-pruned baseline with the same effective depth as TaH (after pruning) could be provided ... isolate the effect of the selective iteration mechanism itself.
>
> We add the requested baseline: Qwen3-1.7B-Base with the same layer removed as in TaH, trained under identical settings as ablations.
>
> |Method|AIME25|Olympiad|AMC23|MATH500|GSM8K|Average|
> |:--|:-:|:-:|:-:|:-:|:-:|:-:|
> |Standard|13.3|33.0|42.2|68.4|82.1|47.8|
> |Standard-Pruned|11.7|32.7|41.3|68.0|79.4|46.6|
> |TaH|17.9|38.8|48.4|74.4|84.5|52.8|
>
> As expected, pruning alone degrades performance. The large gain of TaH over Standard-Pruned (52.8 vs 46.6) clearly isolates the benefit of the selective iteration mechanism.

---

> > ### Author Rebuttal · Reviewer_yJ5W · 2026-04-03
> >
> > I will maintain my postive score

---

> > > ### Author Response · Authors · 2026-04-05
> > >
> > > Thank you for your time and thoughtful engagement. We sincerely appreciate your consistent recognition of our work throughout the review and rebuttal process, and we are glad that our rebuttal has fully addressed all your concerns.

---

### Official Review · Reviewer_2jtn · 2026-03-23

**Soundness:** 4
**Presentation:** 3
**Significance:** 3
**Originality:** 3
**Overall Recommendation:** 5
**Confidence:** 4

**Summary:**

While prior works have proposed looped transformers to improve latent reasoning while preserving parameters count, this paper identifies latent overthinking in them, where the models switch their already correct predictions to wrong predictions with more iterations. Motivated by this observation, the paper proposes selective iteration to improve latent reasoning, and introduces Think-at-Hard (TaH), a looped transformer optimized to support this selective iteration. The recipe provided for training such model is to have an oracle policy with an LLM backbone that is used to determine ground-truth iteration depths. Using that, the paper decouples optimizing computation under the oracle policy's iteration schedule and training an iteration decider that mimics the oracle policy. For the first part, it also introduces duo-causal attention mechanism that enables cross-layer attention. Finally, the paper shows that across several benchmarks, TaH improves accuracy over baselines with minimal overhead by enabling selective iteration.

**Compliance With Llm Reviewing Policy:**

Affirmed.

**Key Questions For Authors:**

- Why are the experiments limited to depth 2? If further iterations do not help, what is the reason? What is the barrier to scale the selective iteration approach to more depths?

- The paper claims "Some previous work supervises all iteration depths with next-token labels, while TaH only supervises final output tokens." Could you please elaborate that? Doesn't TaH fine-tune for NTP across all tokens?

- Could you please explain how you measure Decoding computation (GFLOPs) and memory access (GB)?

**Limitations:**

yes

**Strengths And Weaknesses:**

### Strengths
**Significance**: The paper identifies an important problem. While looping transformers helps with latent reasoning, it might also hurt by overthinking. Therefore, the paper proposes a clever approach, which is selective iteration and provides a practical and creative recipe for training a model optimized for selective iteration. This novel method outperforms the strong baselines, showing its significance.

**Originality**: Designing a selective iterative method brings many challenges that the paper carefully addresses. It also comes up with novel approaches such as due attention mechanism to fill the gaps. This adds to the originality of the paper and makes it a valuable contribution to the literature.

### Weaknesses
**Narrow Practicality**: Despite the novel approach, the experiments being limited to depth 2 makes it less impressive. In order to introduce selective iteration as a new approach, we need to scale the number of iterations further, going beyond just one refining iteration.

---

> ### Author Rebuttal · Authors · 2026-03-31
>
> We sincerely thank Reviewer 2jtn for the strong recognition. We appreciate your valuable appreciation of our **significance, originality, and experiments**. We address the concerns below.
>
> ---
> ### W1 & Q1: Experiments on iteration depth>2
>
> > the experiments being limited to depth 2 makes it less impressive ... scale the number of iterations further, going beyond just one refining iteration.
> > Why are the experiments limited to depth 2? If further iterations do not help, what is the reason? What is the barrier to scale ...?
>
> We expand iteration depths to 3 in Appendix A.2.4, where TaH-3 additionally improves by ~1% over TaH-2 with only ~1% depth-3 iterations. Additional details (including TaH-4) are in our response to Reviewer RduM, W2.
>
> Currently, the main barrier for further scaling is the corresponding infrastructure and system. But note that the dominant untapped gain TaH highlights is to improve iteration accuracy (up to 32% with a perfect oracle in Table 1), not depth. This paper therefore focuses on selective iteration design while being fully compatible with depth scaling.
>
> ---
> ### Q2: Supervision scope: across all tokens?
>
> > The paper claims "Some previous work supervises all iteration depths with next-token labels, while TaH only supervises final output tokens." Could you please elaborate that? Doesn't TaH fine-tune for NTP across all tokens?
>
> Yes, TaH fine-tunes for NTP across **all response tokens**, as in standard SFT. The distinction is about iteration depth: for each token, TaH supervises only the **final iteration** output with the NTP loss, whereas some prior works [1] apply (weighted) NTP loss at **every iteration depth**, including intermediate latent iterations. We will clarify this in the final version.
>
> [1] Zhu, et al. "Scaling latent reasoning via looped language models." arXiv'25.
>
> ---
> ### Q3: How to compute GFLOPs and memory access?
>
> > Could you please explain how you measure Decoding computation (GFLOPs) and memory access (GB)?
>
> We use **operator-level symbolic calculation** in Appendix A.2.3, with wall-clock latency in Appendix A.2.2.
> Specifically, we first collect the real-world statistics (input/output length, iteration depths) in Table 11.
> Then, we symbolically compute costs with the statistics across all neural modules, including linear layers, attention, RMSNorm, RoPE, residual additions, logit-weighted embedding update, and so on, following prior work [2].
> We will add detailed formulations in the revision; the main formulas are below.
>
> [2] Yang, et al. "Glitches: GPU-FPGA LLM inference through a collaborative heterogeneous system." HPEC'24.
>
> ---
> **Notation**
>
> |**Symbol**|**Description**|**Symbol**|**Description**|
> |---|---|---|---|
> |$h$|Hidden dim|$n_h$|Query head count|
> |$s$|KV cache length|$n_{kv}$|KV head count|
> |$d$|Iteration depth|$h_h$|Per-head dim ($h/n_h$)|
> |$l$|Number of layers|$h_{kv}$|Total KV dim ($n_{kv} h_h$)|
> |$w$|Weight count per layer|$h_{ff}$|FFN intermediate size|
> |$b$|Bytes per element|$t_i$|Input token count|
> |$v$|Vocabulary size|$t_o$|Output token count|
> |$k$|Top-$k$ logits|||
>
> **Per-operator, per-layer, and per-step costs**
>
> |**Component**|**FLOPs**|**Memory (bytes)**|
> |---|---|---|
> |*Operators*|||
> |nn.linear|$2h_\text{in}h_\text{out}$|$(h_\text{in}h_\text{out} + h_\text{in} + h_\text{out})b$|
> |nn.sdpa|$4sh$|$[2h + s(2h_{kv} + 3n_h)]b$|
> |*Modules (per layer)*|||
> |FFN|$6hh_{ff}$|$3hh_{ff}b + O(hb)$|
> |Self-Attention|$4h(h+h_{kv}) + 4sh$|$[2h(h+h_{kv}) + s(2h_{kv} + 3n_h)]b + O(hb)$|
> |*Decode step* ($\times l$ layers)|||
> |Step|$l[4h(h+h_{kv}) + 4sh + 6hh_{ff}] + O(lh_{ff})$|$l[w + s(2h_{kv} + 3n_h)]b + O(lhb)$|
>
> Lightweight vector operations (RMSNorm, RoPE, SiLU, residual add) are profiled but omitted above for brevity.
>
> **Standard decoding.**
> The total FLOPs or memory access, denoted by cost $C$, are:
>
> $$C = \sum_{i=1}^{t_o} C_\text{step}(t_i+i-1)$$
>
> **TaH.**
> Each output token $i$ has depth $d_i$, requiring $d_i$ backbone passes, $d_i - 1$ inner transitions, and $\min(d_i, d_\text{max}-1)$ decider calls:
>
> $$ \sum_{i=1}^{t_o} \left[ \sum_{d=1}^{d_i} C_\text{step}\big(s_i^{(d)}\big) + (d_i{-}1) C_\text{trans} + \min(d_i,d_\text{max}{-}1) C_\text{dec} \right] $$
>
> Here, $s_i^{(d)}$ is the KV cache length visible at depth $d$ for token $i$.
> Under duo-causal attention ($d_\text{max}=2$):
> - $s_i^{(1)} = t_\text{in} + i - 1$, since all prior tokens pass through depth 1.
> - $s_i^{(2)} = s_i^{(1)} + |\{j < i : d_j = 2\}|$, consisting of token count plus prior depth-2 tokens.
>
> So overhead is dominated by average depth $$\bar{d} = \frac{1}{t_\text{out}}\sum_i d_i,$$
> giving approximately $\bar{d}\times$ the standard cost when $\bar{d}$ is small.
>
> **AlwaysThink.**
> With $d_i=2$ for all tokens, the two passes incur $2\times$ FFN cost and $\sim 3\times$ attention cost, totaling $2$--$3\times$ overhead.
>
> **Results**
> With real-world statistics, TaH incurs only $1.04$–$1.05\times$ the Standard baseline cost, while AlwaysThink requires $2.19$–$2.27\times$.

---

> > ### Author Rebuttal · Reviewer_2jtn · 2026-04-06
> >
> > Thank you for the detailed rebuttal and answering my questions.

---

### Decision · Program_Chairs · 2026-04-30

**Decision:**

Accept (regular)

**Comment:**

This paper addresses the inefficiency of latent iterations in looped transformers by identifying a "latent overthinking" phenomenon, where initially correct token predictions are erroneously revised in later passes. To solve this, the authors propose Think-at-Hard (TaH), an architecture that utilizes a lightweight neural decider and depth-aware attention mechanisms to dynamically restrict extra computational iterations strictly to difficult tokens.

The reviewers recognized the clear problem formulation, efficient architecture design, and strong empirical results. The concerns from the reviewers were resolved in the rebuttal.